# Balancing true and false detection of intermittent sensory targets by adjusting the inputs to the evidence accumulation process

Anna C Geuzebroek[1]*, Hannah Craddock[1,2], Redmond G O'Connell[3], Simon P Kelly[1]*

[1]School of Electrical and Electronic Engineering and UCD Centre for Biomedical Engineering, University College Dublin, Dublin, Ireland; [2]Department of Statistics, University of Warwick, Warwick, United Kingdom; [3]Trinity College Institute of Neuroscience and School of Psychology, Trinity College Dublin, Dublin, Ireland

**Abstract** Decisions about noisy stimuli are widely understood to be made by accumulating evidence up to a decision bound that can be adjusted according to task demands. However, relatively little is known about how such mechanisms operate in continuous monitoring contexts requiring intermittent target detection. Here, we examined neural decision processes underlying detection of 1 s coherence targets within continuous random dot motion, and how they are adjusted across contexts with weak, strong, or randomly mixed weak/strong targets. Our prediction was that decision bounds would be set lower when weak targets are more prevalent. Behavioural hit and false alarm rate patterns were consistent with this, and were well captured by a bound-adjustable leaky accumulator model. However, beta-band EEG signatures of motor preparation contradicted this, instead indicating lower bounds in the strong-target context. We thus tested two alternative models in which decision-bound dynamics were constrained directly by beta measurements, respectively, featuring leaky accumulation with adjustable leak, and non-leaky accumulation of evidence referenced to an adjustable sensory-level criterion. We found that the latter model best explained both behaviour and neural dynamics, highlighting novel means of decision policy regulation and the value of neurally informed modelling.

*For correspondence:
anna.geuzebroek@ucd.ie (ACG);
simon.kelly@ucd.ie (SPK)

## Editor's evaluation

This important work reveals a novel mechanism by which perceptual decision-making is regulated as a function of task demands. The combination of behavioral and physiological (EEG) evidence supporting the accumulation of evidence referenced to a context-dependent sensory criterion is convincing. Overall, the study makes a strong case for the importance of augmenting behavioral modeling with additional input from neural signatures of the underlying decision process.

## Introduction

The way that we form decisions is highly adaptable to task demands. It is now widely understood that making accurate decisions on noisy sensory information involves the accumulation of evidence up to a criterion amount or 'bound' (*Gold and Shadlen, 2007*; *Ratcliff and Smith, 2004*). This bound is seen as the primary means to strategically adapt perceptual decision policies to meet various task demands. Significant behavioural and neural evidence has now amassed to establish that bound

adjustments are indeed carried out for certain, well-studied scenarios, such as when trading speed for accuracy in discrete, two-alternative forced choice decisions (*Bogacz et al., 2010*; *Palmer et al., 2005*), but it is not clear whether this mechanism generalises to all situations that require policy adaptations. In particular, in the pervasive daily life scenario in which unpredictably timed targets must be detected within continuous streams of noisy sensory information, one's decision policy must be tailored to rapidly detect as many targets as possible while keeping false alarms to a minimum (*Lasley and Cohn, 1981*). Here, an overly liberal policy could result in an excessive amount of false alarms driven solely by noise in between targets, while an overly conservative policy will miss too many targets (*Gold and Stocker, 2017*; *Ossmy et al., 2013*). In such contexts, expected target strength is a key factor in setting a decision policy to optimise performance – contexts with weaker targets call for a more liberal decision policy. Although bound adjustments present a natural mechanism for these policy adjustments, this has never been experimentally tested for this scenario.

Much has been learned about the neural implementation of decision-bound adjustments from studies of the neural basis of the speed-accuracy trade-off. In these studies, motor-selective cortical networks have been shown to implement an increase of baseline activity prior to evidence onset, emphasising speed by effectively decreasing the criterion amount of cumulative evidence needed to reach an action-triggering threshold (*Forstmann et al., 2008*; *Forstmann et al., 2010*; *Ivanoff et al., 2008*; *van Veen et al., 2008*; *Wenzlaff et al., 2011*). Additionally, recent behavioural and neurophysiological findings have established the operation of a time-dependent 'urgency' signal in these motor circuits, an evidence-independent build-up component that drives the decision process closer to a fixed decision threshold, gradually decreasing the cumulative evidence required to respond (*Churchland et al., 2008*; *Hanks et al., 2014*; *Murphy et al., 2016*; *Thura and Cisek, 2016*). However, little is known about how such mechanisms may participate in continuous detection scenarios, or in the adjustments needed to account for other task-related factors such as the expected target evidence strength. Our initial prediction for the present study was that the decision bound – that is, the criterion amount of cumulative evidence – would be lower in continuous detection contexts containing weaker targets. However, in continuous detection where target onsets are unpredictable, the decision process must operate continuously, offering other parameters that could alternatively be adjusted. For example, if accumulation proceeds continuously it is unlikely to be perfect because this generates excessive false alarms (*Ossmy et al., 2013*) and would instead entail some form of information loss such as through non-linear or leaky accumulation, which has been shown to form a critical adjustable component governing decision-making policies in volatile environments (for review, see *Gold and Stocker, 2017*; *Glaze et al., 2015*; *Harun et al., 2020*; *Murphy et al., 2021*).

Our findings indicate that observers can adjust their decision-making policy based on their knowledge of the difficulty context, but not by adjusting bound settings in the way expected from observed speed/accuracy adjustments. While the electrophysiological signature of motor preparation gradually approached its action-triggering threshold through the inter-target interval (ITI) consistent with the operation of evidence-independent urgency, this baseline activity was not set higher in weak target contexts in order to reduce the bound as predicted. Instead, the motor preparation signal suggested that the bound was reduced in the strong context. We therefore hypothesised that an alternative way to set a conservative decision policy is to reduce the information gathered in the accumulator, and tested two ways of implementing this: (1) leaky accumulation with context-dependent leak and (2) a non-leaky accumulation of evidence referenced to a context-dependent sensory criterion. Fitting these models with urgency dynamics directly constrained by the observed motor preparation signals, we found that the latter model produced a superior fit to both behavioural data and the dynamics of a motor-independent signature of evidence accumulation, the centro-parietal positivity (CPP; *Kelly and O'Connell, 2013*; *O'Connell et al., 2012*). We are thus able to falsify the initially predicted bound-adjustment model in favour of an account based on the adjustment of a sensory-level criterion that regulates the transfer of incoming evidence into the accumulation process.

## Results

### Behavioural signatures of contextual adjustments

We analysed the data of 14 participants that continuously monitored a cloud of randomly moving dots for unpredictable, intermittent targets defined by coherent upward motion. Within each block, the

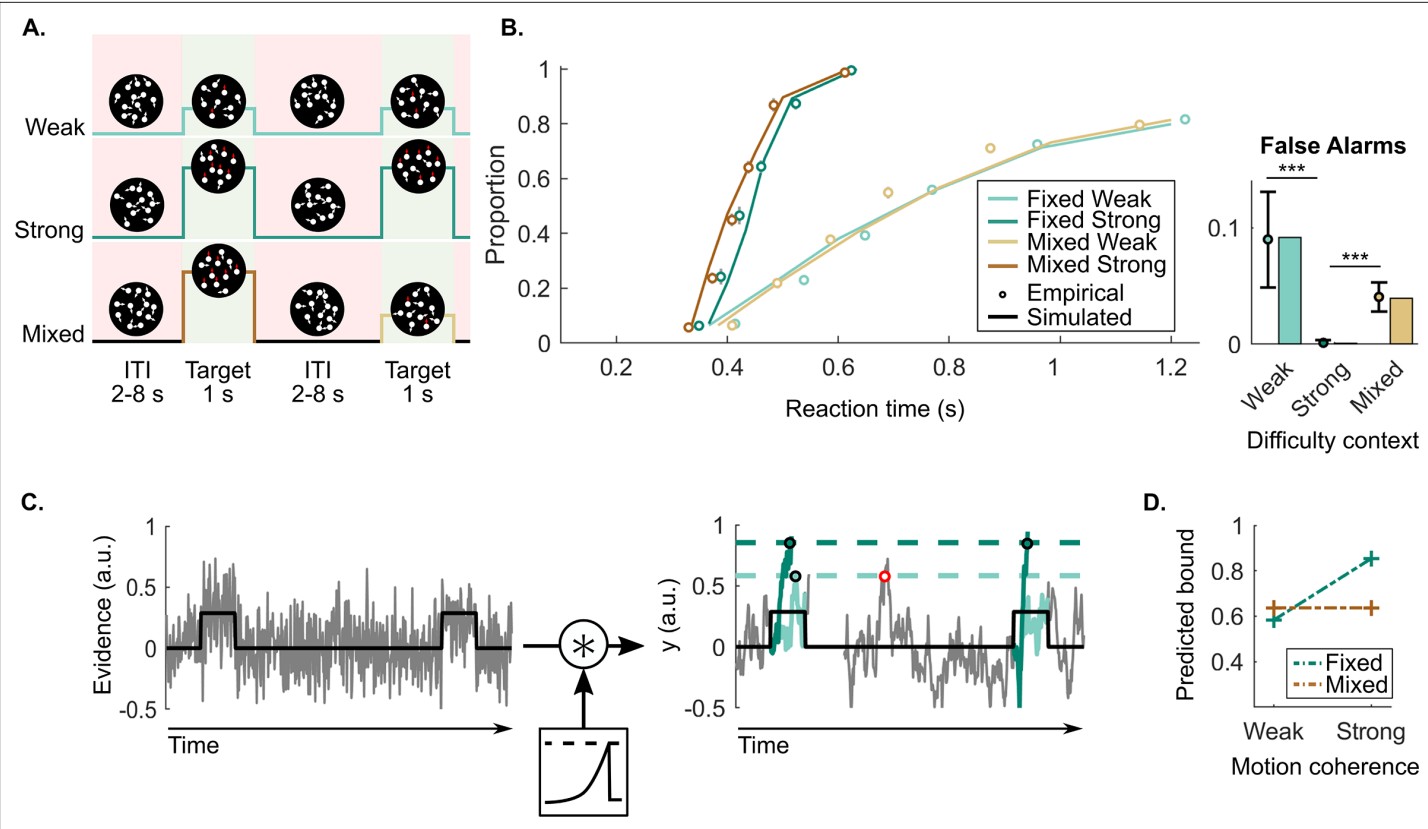

**Figure 1.** Continuous random dot motion (RDM) detection task and behaviour. (**A**) Participants continuously monitored a cloud of moving dots to detect targets defined by a step change to coherent upward motion for 1 s. The inter-target interval (ITI) duration varied between 2, 4, 6, and 8 s. In each block of 24 targets, participants would perform a weak fixed context condition (step increase of 25%), a strong fixed context condition (step increase of 70%), or a mixed context condition (25% and 70% equally likely). (**B**) Reaction time (RT) cumulative quantile probability functions and the proportions of a 2 s ITI period containing a false alarm as a function of difficulty context (dots in cumulative quantile probability and false alarm plots). Each plot includes simulated data from a fitted leaky accumulator model with bound adjustment (solid lines in cumulative quantile probability plot and bars in the false alarm plot). (**C**) Schematic representation of the leaky accumulator model with adjustable bound parameters. Noisy evidence, which steps up during targets, is accumulated ($y(t)$) until it reaches a context-dependent bound. Leaky accumulation is represented as a convolution with an exponential decay kernel. The degree of leakage is a free parameter but constrained to be equal (not adjusted) across contexts. When the target evidence is weak, a more liberal (lower) bound needs to be set to avoid misses; this comes however at the cost of more false alarms (example indicated in red). (**D**) Bound parameter values estimated by the best fitting bound-adjustment model. All figures represent a sample size of N = 14. Error bars represent the 95% confidence interval. p-Values resulting from the GMM are indicated by the asterisks; *p<0.05, **p<0.01, and ***p<0.001.

difficulty context was set to be fixed with (1) only weak targets, (2) only strong targets, or (3) mixed with both randomly occurring weak and strong targets. The mixed context was included to control for differences due to target motion coherence itself as distinct from context (see **Figure 1A**). As expected, participants detected targets more accurately and faster when the targets were strong for both the fixed and mixed context (**Figure 1B**; main effect of motion coherence on hit rates in generalised mixed regression model (*GMM*), $\chi^2(1)=8.5$, p<0.001, and reaction time (RT) in repeated-measures ANOVA (*rANOVA*), F(1, 13)=371.8, p<0.001). Additionally, participants were faster in the mixed context than the fixed ones (main effect of difficulty context on RT in *rANOVA*, F(1, 13)=21.3, p<0.001).

False alarm rates demonstrate a qualitative signature of contextual adaptations, with approximately zero in the strong context, higher false alarms rates in the weak context probability of 0.09 per 2 s ITI period, and intermediate rates in the mixed context (probability of 0.04 per 2 s ITI period; **Figure 1B**; *GMM*, $\chi^2(1)=5.7$, p<0.001 and $\chi^2(1)=5.5$, p<0.001, respectively). In accumulator models of decision making, these patterns can be captured by setting a higher bound when the task is easy, to avoid false alarms while still detecting all targets (see **Figure 1C**). Confirming this, a leaky accumulator model with a higher bound for the strong context (**Figure 1D** and **Table 1** for all the fitted parameters) was able

**Table 1.** Parameter values estimated for the leaky accumulator model with adjustable bound fit to behaviour alone, and for the neurally informed models featuring leak adjustment and criterion adjustment. 'Noise' refers to the standard deviation ($\sigma$) for the Gaussian sensory evidence noise. 'Bound' refers to the threshold set on the decision variable for triggering a response. 'Leak' refers to the proportion of the current cumulative evidence total that leaks away on the following sample (note a 16.7 ms time step is used). No leak was fitted for the criterion-adjustment model. 'Tnd' refers to the non-decision time in ms. 'Drift' refers to drift rate, corresponding to the mean of the sensory evidence during targets. The goodness-of-fit metric, $G^2$, is listed for each model. Note that, if comparing parameter values between models directly, it must be taken into account that whereas the bound-adjustment model set a scaling parameter of noise $\sigma = 0.1$ and allowed bounds to vary freely with respect to this. In contrast, the neurally constrained models were scaled directly by the normalised urgency signals relative to the ultimate action-triggering bound taken to be equal to 1. These fixed parameters are indicated in red.

| | Noise | Variable parameters | | | | Tnd | Drift | | | | $G^2$ |
|---|---|---|---|---|---|---|---|---|---|---|---|
| | | Bound | | | Leak | | | | | | |
| Bound-adjustment | | W | S | M | | | W | S | MW | MS | |
| | 0.1 | 0.58 | 0.85 | 0.63 | 0.14 | 238 | 0.05 | 0.14 | 0.05 | 0.12 | 16 |
| | | Leak | | | Bound | | | | | | |
| Leak-adjustment | | W | S | M | | | W | S | MW | MS | |
| | 0.05 | 0.02 | 0.08 | 0.04 | 1 | 238 | 0.02 | 0.06 | 0.03 | 0.07 | 23 |
| | | Criterion | | | | | | | | | |
| Criterion-adjustment | | W | S | M | | | W | S | MW | MS | |
| | 0.07 | 0.02 | 0.06 | 0.03 | 1 | 230 | 0.03 | 0.10 | 0.04 | 0.09 | 16 |

to accurately reproduce behavioural data, particularly the important context-dependent false alarm pattern (see bars of the false alarm plot in *Figure 1B*; $G^2$=16).

## EEG signatures of decision formation: adjustments across contexts

To test for electrophysiological evidence for the putative bound adjustment, we examined two signals known to reflect the dynamics of decision formation: decreases in beta frequency band activity (15–30 Hz) over motor cortex contralateral to the movement, reflecting motor preparation (see *Figure 2A*), and the CPP, reflecting motor-independent evidence accumulation (see *Figure 2D*).

It has long been established that beta oscillatory activity over motor cortical EEG sites gradually decreases in amplitude as participants prepare to make a movement with their contralateral limb (*Donoghue et al., 1998*; *Pfurtscheller et al., 1996*). More recently, several replicating findings further indicate that this beta-indexed motor preparation signal reflects a build-to-threshold decision variable driven by a combination of evidence accumulation and urgency. First, it builds at an evidence-dependent rate during decision formation and reaches a stereotyped threshold level at the time of motor execution (*Kelly et al., 2021*; *O'Connell et al., 2012*). Second, fluctuations in pre-target beta activity have been shown to predict trial-to-trial behavioural variability in the way accumulator starting point variability does in models (*de Lange et al., 2013*; *Donner et al., 2009*; *Gould et al., 2012*). Third, pre-target beta is systematically shifted closer to its pre-response threshold level under conditions of speed emphasis (*Murphy et al., 2016*; *Steinemann et al., 2018*; *Kelly et al., 2021*). Fourth, in scenarios where evidence onset is anticipated, beta starts building towards its threshold even before evidence onsets. This is consistent with a dynamic urgency component and aligns with other behavioural and electrophysiological markers of dynamic urgency observed in the similar scenarios (*Kelly et al., 2021*; *Corbett et al., 2023*).

In line with this body of previous research, we took the amplitude reached by beta 80 ms prior to response (allowing for motor execution delays) to correspond to the threshold level required to trigger a response in a given condition. This way, the level in the pre-target baseline period can be taken to

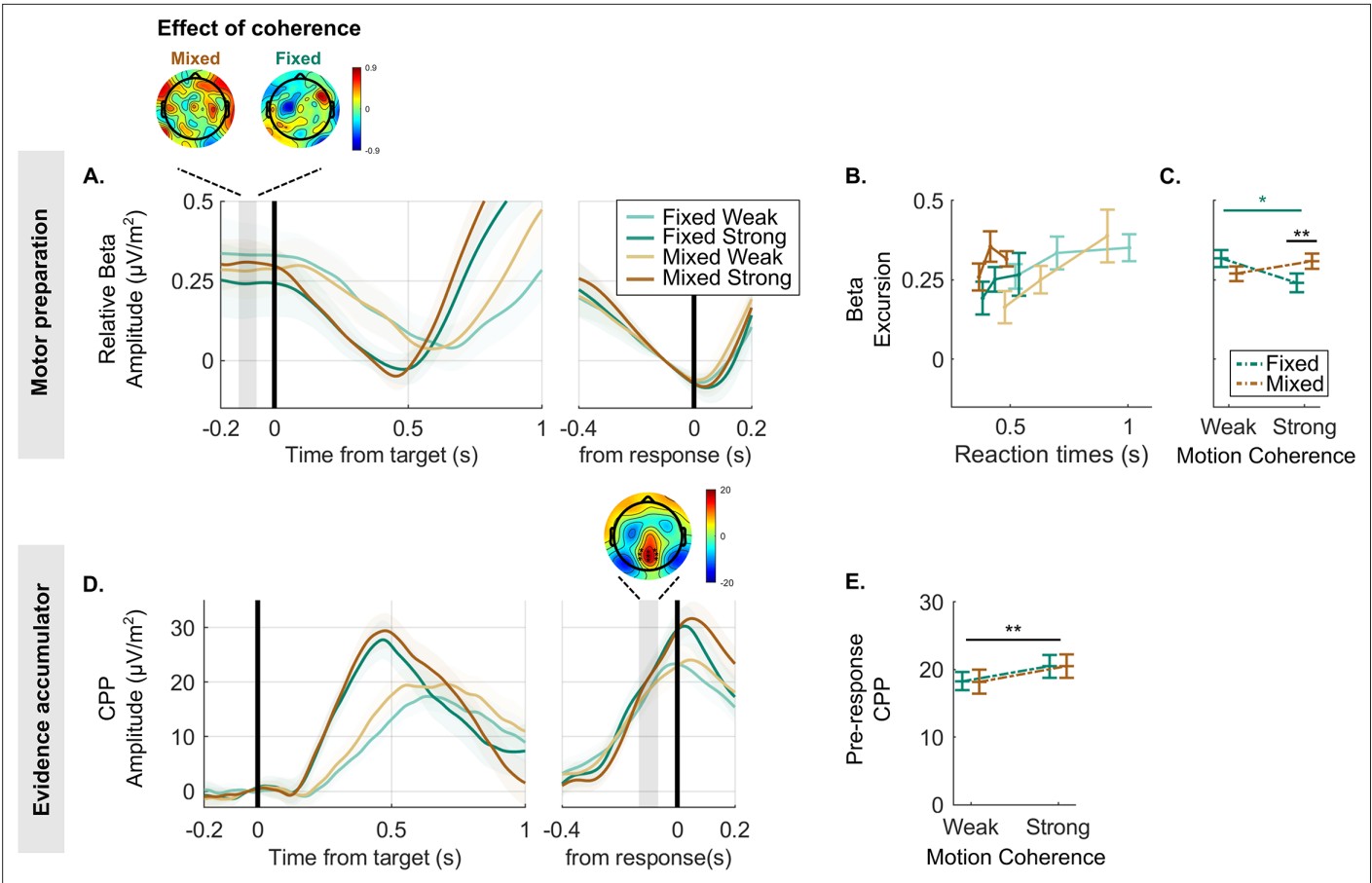

**Figure 2.** Decreases in beta amplitude track motor preparation and reveal a lower decision bound for the strong context. (**A**) Beta amplitude (15–30 Hz) over the motor cortex, aligned to the target (left) and to the response (right), averaged for each condition and expressed as a difference relative to the corresponding pre-response level taken to reflect motor execution threshold (i.e. threshold level is indicated by zero). To ensure that the beta rebound effect following false alarms (see **Figure 2—figure supplement 1**) could not artificially drive the differences in excursion across conditions, targets occurring within 1.2 s of a false alarm were excluded from these data. The scalp topography shows the distribution of the difference between weak and strong targets in the baseline amplitude (again relative to the motor execution threshold; blue corresponds with more negative amplitude for strong relative to weak motion coherence). This highlights that the difference in beta excursion in the fixed strong relative to fixed weak context is maximal over the motor cortex. (**B**) The average beta excursion (i.e. difference of pre-target beta amplitude minus pre-response beta amplitude), plotted for three equal-size reaction time (RT) bins per condition. Smaller beta excursion predicts faster RTs as found in previous studies linking beta to a motor-level decision variable. (**C**) Grand-average beta excursion for all conditions, reflecting the neural index of bound settings. These show an obvious qualitative difference to the bound parameter values estimated in the behavioural data fit of the leaky accumulator model with adjustable bound in **Figure 1D**. (**D**) Centro-parietal positivity (CPP) signals averaged for each condition, target-locked (left) and response-locked (right). The topography of the indicated pre-response time window shows the cluster of electrodes used for plotting average waveforms. (**E**) Average pre-response CPP amplitude for each condition. All figures represent a sample size of N = 14. Error bars represent the 95% confidence interval. p-Values resulting from the GMM are indicated by the asterisks; *p<0.05, **p<0.01, and ***p<0.001.

The online version of this article includes the following figure supplement(s) for figure 2:

**Figure supplement 1.** Beta waveforms aligned to (**A**) a true detection and to (**B**) a false alarm, both plotted as a function of context (weak, strong, and mixed).

**Figure supplement 2.** Posterior N2 component.

represent the decision variable starting point at the time of evidence onset. Thus, the difference between these two, known as the 'excursion', indexes the change in motor preparation required to commit to a detection decision and thus indexes bound settings in a given context (**Heitz and Schall, 2012**; **Kelly et al., 2021**, see below for separate analysis of baseline and pre-response amplitudes). First, to support the use of beta activity to reflect the motor-level decision variable in this dataset, we confirmed that it built up more steeply during higher coherence targets (**Figure 2A**; GMM, $\chi^2(1)=-2.8$, p=0.004), and that faster RTs within a given condition were associated with the signal starting

relatively closer to pre-response threshold (*Figure 2B*; GMM, $\chi^2(1)$=2.6, p=0.008). Additionally, we found that the beta amplitude prior to a false alarm closely matched to the amplitude reached prior a true target detection, verifying the assumption that this threshold level is applied consistently within each block (see *Figure 2—figure supplement 1*). Turning then to context-dependent adjustments, we found that difficulty context (fixed vs. mixed) and motion coherence (weak vs. strong) had a significant interactive effect on beta excursion (*Figure 2C*; GMM, $\chi^2(1)$=2.8, p=0.006). Contrary to the prediction of the bound-adjustment model (*Figure 1D*), this was driven by a pattern whereby beta excursion was smaller, not larger, in the strong context relative to other conditions, indicative of a lower decision bound (weak vs. strong in fixed condition: $\chi^2(1)$=–2.3, p=0.02; strong fixed vs. strong mixed: $\chi^2(1)$=2.1, p=0.03).

More recently, another signature of decision formation has been characterised – the CPP – which, like beta, builds at an evidence-dependent rate to a peak around the time of decision commitment and response (*O'Connell et al., 2012*; *Kelly and O'Connell, 2013*). This signal, unlike beta, builds even in the absence of motor requirements, but does not undergo baseline shifts according to speed/accuracy emphasis (*Steinemann et al., 2018*). The current working hypothesis is thus that the CPP reflects pure cumulative evidence which feeds into motor-level decision variable alongside other evidence-independent signal components such as urgency. In support of this scheme, simulations of pure cumulative evidence build-up have been shown to match empirical CPP waveforms in their dynamics and pre-response amplitude patterns across conditions and RTs (*Afacan-Seref et al., 2018*; *Kelly et al., 2021*; *Twomey et al., 2015*). Here, a hypothetical higher bound setting in the strong context would predict that more cumulative evidence is required before reaching the movement execution threshold, and hence a higher CPP amplitude. More specifically, a true contextual adjustment effect should give rise to a CPP amplitude difference in the strong relative to weak context

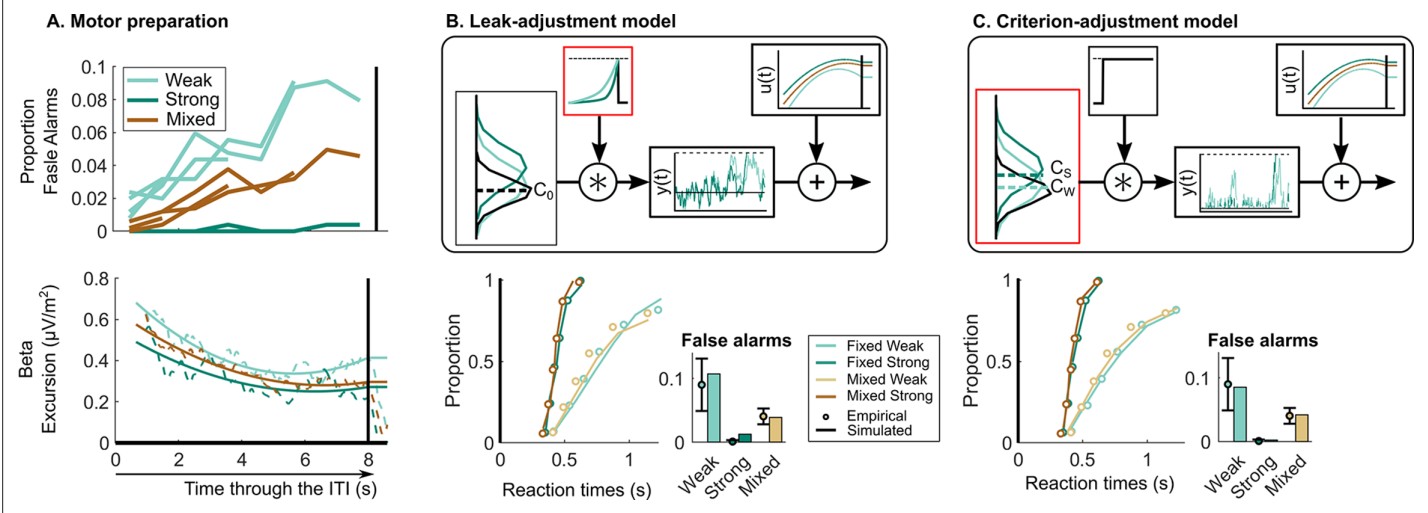

**Figure 3.** Neurally informed modelling. (**A**) Motor preparation throughout the inter-target interval (ITI), showing a dynamic urgency trend. The proportion of false alarms (per 1 s period) increases throughout the ITI (*upper*; four ITIs superimposed). This corresponds to an increase in motor preparation throughout the ITI reflected in decreasing beta amplitude (*lower*). Beta amplitude is plotted for the 8 s ITI relative to the corresponding pre-response level taken to reflect motor execution threshold. After normalisation, beta amplitude is used to model urgency ($u(t)$) through the ITI generating the observed increase in false alarms. Model evaluation for the neurally informed (**B**) leak-adjustment and (**C**) sensory criterion-adjustment model comparing empirical and predicted reaction time (RT) cumulative quantile probability distributions and false alarm rates (per 2 s ITI period). Above are model schematics illustrating the different mechanisms of continuous accumulation and context-dependent adjustment. In (**B**) the leak-adjustment model, evidence is referenced to the centre of the noise distribution ($C_0$, equal to zero) and accumulated ($y(t)$) with a leak. Leak is represented in the time constant of a convolution kernel and is free to change across contexts. Meanwhile in (**C**) the criterion-adjustment model, evidence is referenced to a criterion 'zero' so that is adjustable across contexts ($C_W$, $C_S$, and $C_M$ with two shown for illustration) and fed to a non-leaky accumulator with a lower reflecting bound at zero to preclude negative accumulation. All figures represent the empirical data of N = 14. Error bars represent the 95% confidence interval.

The online version of this article includes the following figure supplement(s) for figure 3:

**Figure supplement 1.** Neurally informed models fit simultaneously to behaviour and beta excursion.

**Figure supplement 2.** Simulated single-trial example of the motor-level decision variable ($y(t) + u(t)$) during the inter-target interval (ITI).

over and above any difference between high and low coherence targets in the mixed context. This, however, was not observed in the empirical pre-response CPP, where only a significant motion coherence effect was found (*GMM*: $\chi^2$(1)=5, p<0.001; *Figure 2D and E*). Tracing the waveforms beyond the pre-response measurement window shows that the CPP tends to climb higher for the mixed strong than fixed strong condition, if anything (*Figure 2D*). The electrophysiological data are therefore inconsistent with the bound-adjustment model. This raises the question of whether an alternative mechanism can explain the behavioural markers of policy adjustment while also producing the observed dynamics in neural decision signals.

## Alternative neurally informed models

Following a recent approach (*Kelly et al., 2021*), we constructed models whose decision-bound settings were constrained to match the observed beta excursion measurements. We use beta activity as it best corresponds to the motor-level decision variable which is ultimately subjected to a threshold. After fitting to behaviour, we then examined whether key dynamical features of evidence accumulation predicted by the models matched those of the observed CPP time courses in a further validation step. To determine the beta activity-based model constraints, it was important to first characterise the dynamics of motor preparation during the ITI more fully. Beta waveforms during the full ITIs showed that in addition to the static shift towards the motor execution threshold in the strong context, there was a strong tendency for beta to decrease over the course of the ITI in all contexts (*Figure 3A*, *lower*). This increasing trend in beta mirrored a tendency for false alarm rate to increase during the ITI (*Figure 3A*, *upper*). In keeping with our recent work (*Corbett et al., 2021*; *Kelly et al., 2021*), we assumed that beta-indexed motor preparation is driven by the sum of evidence accumulation and evidence-independent urgency (*Churchland et al., 2008*; *O'Connell et al., 2018*), and reasoned that since evidence is zero mean during the ITI, beta dynamics can be taken to index urgency alone during that period. Accordingly, we took the grand-average beta amplitude waveform during the 8 s ITI and baseline-corrected these relative to the motor execution threshold indexed by average pre-responses in a given context. These traces were normalised and flipped so that 1 corresponded to the motor execution threshold level and 0 to the lowest level of motor preparation during the ITI. To this waveform a second-order polynomial was fitted to capture the smooth urgency trend relative to threshold (see Materials and methods for more details). This signal was added to the output of the continuous evidence accumulation and the resultant decision variable was subjected to the ultimate, constant decision threshold (see *Figure 3B and C*).

Using these constraints, we explored other possible mechanisms of continuous accumulation and context-dependent adjustment. Since, contrary to expectations, the offsets in beta waveforms during the ITI would on their own predict increased false alarms in the fixed strong condition, there must be another adjustment applied elsewhere to counteract this. First, we considered that the same leaky accumulation mechanism as the bound-adjustment model may be at play but with the adjustment across contexts being applied to leak rather than bound (*Figure 3B*). Here, a more forgetful (higher leak) integration would reduce the risk of false alarms driven by noise (*Glaze et al., 2015*; *Murphy et al., 2021*; *Ossmy et al., 2013*) in a way similar to an increased bound, and thus presents an alternative means of setting a conservative policy. Second, we considered that, alternatively, a context-dependent evidence criterion could be set at each momentary evidence sample serving as a 'zero' reference placed between noise and signal evidence distributions, similar to the 'drift criterion' principle proposed by *Ratcliff, 1985*; *Ratcliff et al., 1999*; *Ratcliff and Tuerlinckx, 2002*. This criterion is distinct in meaning to that of signal detection theory (SDT) in that it does not on its own define the ultimate decision rule to categorise signal versus noise, but rather serves as a reference subtracted from the evidence so that only evidence samples substantially above the noise would be positively accumulated towards the decision bound, while noise alone would conversely tend to count negatively. To prevent runaway negative accumulation in this model, we adopt a reflecting lower bound at zero in the accumulator process (*Usher and McClelland, 2001*). Thus, this last model has two major distinctions: it assumes that an adjustable criterion is set directly on the momentary evidence in addition to the adjustable decision bound on the cumulative evidence (here linked directly to beta excursion), and involves a different form of information loss than leak which nevertheless has a similar effect of reducing risk of false alarms due to noise (see *Figure 3C*). False alarms in this model can

**Table 2.** Parameter values estimated for neurally informed models fitting behaviour and beta excursion simultaneously (see *Figure 3—figure supplement 1*). Neurally informed models were revised to directly constrain the temporal profile of the urgency signal during the inter-target interval (ITI) in the mixed condition only. Two free additional parameters were added for the additive offsets ('shift') in urgency for the weak and strong context condition relative to the mixed context. Additionally, goodness-of-fit metrics are reported as $G^2$+beta penalty.

| | Noise | Variable parameters | | | Tnd | Drift | | | | Shift | | $G^2$ + Penalty |
|---|---|---|---|---|---|---|---|---|---|---|---|---|
| | | Leak | | | | | | | | | | |
| Leak-adjustment | | W | S | M | | W | S | MW | MS | W | S | |
| | 0.05 | 0.03 | 0.08 | 0.04 | 238 | 0.03 | 0.07 | 0.03 | 0.07 | 0.02 | 0.03 | 33 |
| | | Criterion | | | | | | | | | | |
| Criterion-adjustment | | W | S | M | | W | S | MW | MS | W | S | |
| | 0.09 | 0.02 | 0.10 | 0.014 | 248 | 0.04 | 0.14 | 0.05 | 0.10 | −0.14 | 0.06 | 16 |

be prevented by setting a higher evidence criterion, that is creating a stricter reference point distinguishing target evidence samples from noise.

While both neurally informed models mimic the behavioural patterns well (see *Figure 3B and C*, goodness-of-fit in *Table 1* and *Figure 3—figure supplement 2*), the leak-adjustment model provides a poorer fit ($G^2$=23) to the behavioural data than the criterion-adjustment model ($G^2$=16; see *Figure 3B*, *lower*). Remarkably, despite being constrained to adopt the precise bound settings indexed by beta excursion, which oppose the adjustments predicted by the bound-adjustment model, the criterion-adjustment model was able to attain the same goodness-of-fit to behaviour as the initial bound-adjustment model ($G^2$=16).

In the above neurally constrained model fits, we assumed that the cumulative evidence component of motor preparation is negligible during the ITI so that beta excursion directly constrains the urgency component only. A potential issue with this arises from the fact that the accumulator operates continuously in our models: whereas the zero-mean noise in the ITI would accumulate to an average level of zero in the leaky accumulation model, the lower reflecting bound in the criterion-adjustment model would cause a positive asymptotic value of the average evidence accumulation during the ITI which scales inversely with criterion. This would predict highest tonic activity in the weak context as it has the lowest criterion, so it does not constitute a viable alternative explanation the observed lowest beta offset in the weak context. Nevertheless, it highlights a likely inaccuracy in the model's estimation of the degree of adjustment of criterion and urgency offsets. We therefore carried out a revised model fitting procedure in which urgency offsets were not set directly by beta excursion, but rather free to vary in order to reproduce the observed static shifts in the beta excursion across contexts simultaneously with the behavioural data (see detailed explanation in *Figure 3—figure supplement 1*). The behavioural fits still favoured the criterion-adjustment model ($G^2$+penalty = 24) over the leak-adjustment model ($G^2$+penalty = 33; see *Table 2*), and as expected, estimated even larger urgency offsets to compensate for the asymptotic evidence accumulation levels.

To provide further model validation, the empirical CPP was compared against the simulated evidence accumulation time courses of the different models. In the case of the neurally informed models, predicted signatures of bound settings (e.g. cumulative evidence at response) are not informative for validation because they are forced to equal the observed beta excursions by design (see Discussion for comparison of the observed CPP pre-response amplitudes with observed beta excursions). Instead, for the comparison of these two models, respectively featuring leaky accumulation and criterion-referenced, non-leaky accumulation, the stimulus-locked dynamics of accumulator build-up rate (i.e. first derivative) are particularly informative. While a leaky accumulator would be expected to begin to decline in its steepness immediately as it starts to build, a non-leaky accumulator would be expected to rise linearly with a steady slope for a certain amount of time. To test this and compare the models, we analysed the slope profile of the observed and simulated accumulation waveforms. We focused on the strong conditions which will be the most diagnostic in this regard because the strong evidence onset produces a high, robust initial slope, as well as a sharp and relatively invariant CPP

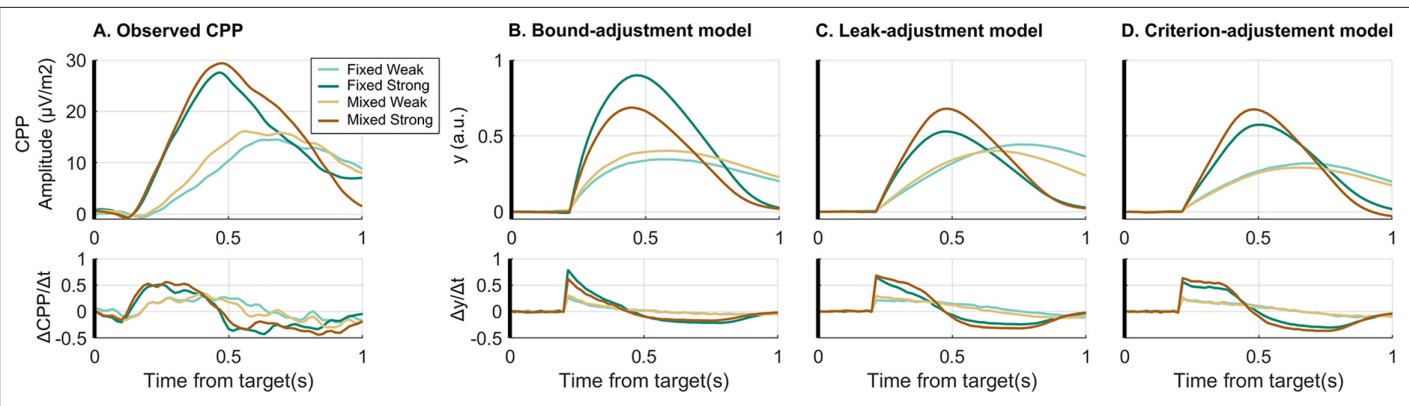

**Figure 4.** Comparison of empirical and simulated evidence accumulation waveforms. (**A**) Empirical average target-locked centro-parietal positivity (CPP) signals (upper row) for each condition as well as their first derivative (lower row). This can be compared to the simulated accumulator process, $y$, for (**B**) the bound-adjustment model, and the neurally informed (**C**) leak-adjustment and (**D**) criterion-adjustment models. The build-up rate of the CPP does not immediately begin to fall steeply in the way predicted by the leaky accumulator model. In these simulations, CPP is simulated as the cumulative evidence without direct urgency influence (***Kelly et al., 2021***). Without loss of generality for the behavioural responses, we assumed that after reaching commitment, there is a delay of 80 ms before the CPP stops accumulating and over 416 ms falls linearly back to zero, implemented identically in all contexts and in both models. This was based on observed post-response dynamics in the real CPP.

The online version of this article includes the following figure supplement(s) for figure 4:

**Figure supplement 1.** Comparison of empirical response-locked centro-parietal positivity (CPP) with the simulated pre-response evidence accumulation waveforms.

build-up onset which best approximates the onset invariance assumed in the models. The fixed strong and, to a lesser degree, the mixed conditions also had stronger predicted degrees of leakage, so that any leak-induced drop in slope should be most apparent against EEG noise. By comparison, the weak conditions have a much less clear onset time presumably due to greater variability in build-up onset timing not accounted for in the model. We found that the empirical CPP slope time course plateaued in the strong conditions indicating a linear build-up profile (***Figure 4A***), whereas the simulated time course of the leaky accumulator models indeed showed a steep drop-off in temporal slope after an initial peak (***Figure 4B/C***). Meanwhile, the criterion-adjustment model with non-leaky accumulation showed a plateau in the slope similar to the empirical CPP (***Figure 4D***). To capture this key feature quantitatively, we computed the percentage drop in slope in the first 100 ms relative to its initial peak. For the fixed strong condition the empirical slope dropped by 5% (bootstrap $CI_{95\%}$ = [–5% 16%]), while the criterion-adjustment model predicted a drop of 6%, and the leak-adjustment model a drop of 27.5%, the latter falling outside the empirical bootstrapped 95% CI. Similarly, in the mixed strong condition the empirical slope dropped by 1.5% (bootstrap $CI_{95\%}$ = [–9.5% 13%]), compared to 3% predicted by the criterion-adjustment model and 17% predicted by the leak-adjustment model.

## Baseline and pre-response beta amplitudes and their relationship with RT

The above analysis of the beta excursion showed that participants began their decision process with a smaller distance to the motor execution threshold in the strong context than for the other conditions (***Figure 2A***). The dynamics of beta amplitude relative to the action-triggering threshold were the key element for constraining the models as these amplitudes dictate the criterion amount of cumulative evidence required to trigger a response. However, to help interpret these changes in terms of context-related differences in psychological state, it is also of interest to separately examine variations in absolute pre-target and pre-response beta amplitudes across and within conditions. At first blush the smaller excursion in the strong context appears consistent with participants being toni-cally more prepared to make a movement, which seems counter-intuitive for a task condition that is by far the easiest. However, pre-target beta amplitude was actually found to be higher, reflecting lower baseline motor preparation in the strong context than both the weak and mixed context (see ***Figure 5A***, *left* and ***Figure 5B***, GMM, $\chi^2(1)=2.3$, p=0.02 and $\chi^2(1)=2.4$, p=0.02, respectively). Mean-while, pre-response beta amplitude was also significantly higher for the strong context, but to an even

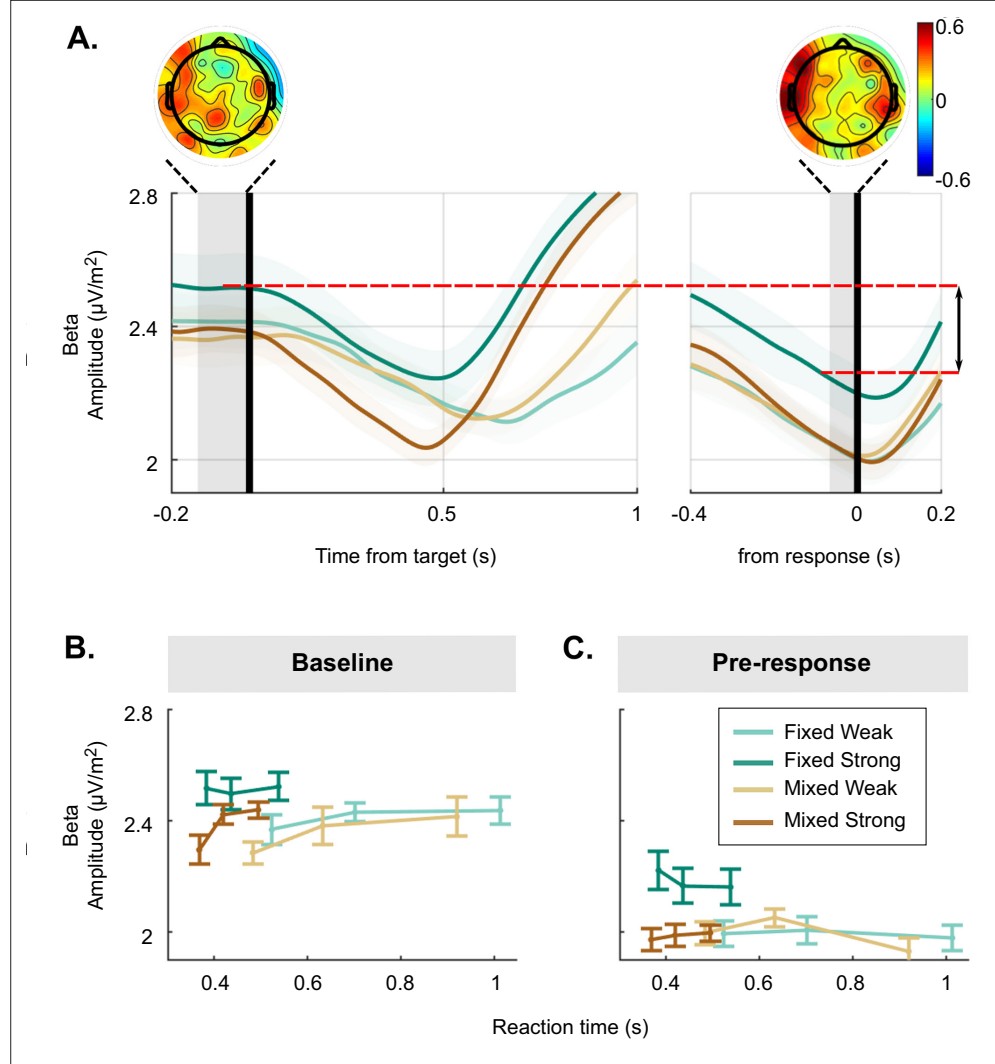

**Figure 5.** Detailed analysis of raw beta amplitude in the baseline and pre-response time frames. (**A**) Beta amplitude (15–30 Hz) over the motor cortex without any baseline or threshold subtraction, aligned to the target (*left*) and to the response (*right*), averaged for each condition. Additionally, a visual representation of excursion is shown with the red lines. Scalp topographies show the distribution of the difference between strong targets in the mixed and fixed condition, more red indicating higher activity in the fixed condition. (**B**) Baseline beta activity and (**C**) pre-response beta plotted for each condition and three equal-size reaction time (RT) bins. Error bars for all plots represent the 95% confidence interval.

greater degree than at baseline (see *Figure 5A*, *right* and *Figure 5C*, GMM, $\chi^2(1)=3$, p=0.003). Thus, compared to the mixed and weak contexts, the much easier strong context appeared to have lower tonic levels of preparation, but also a disproportionately more liberal threshold level, resulting in the smaller beta excursion overall.

We also examined variation of pre-target beta amplitude as a function of RT. As stated above, beta excursion was larger on trials with longer RTs. Interestingly, examining the baseline and pre-response amplitudes separately indicated that this RT relationship was driven by a correlation with beta amplitude in the baseline for the mixed and weak conditions (as seen previously, for example *Steinemann et al., 2018*), whereas it was driven by a correlation with pre-response amplitude in the strong context. This was indicated by a significant interaction between context, motion coherence, and RT for both baseline beta (GMM, $\chi^2(1)=-0.43$, p=0.004) and pre-response beta (GMM, $\chi^2(1)=-0.3$, p=0.007). Post hoc tests showed that baseline beta significantly varied with RT for both weak conditions (both weak conditions p<0.02) and the mixed strong conditions (GMM, $\chi^2(1)=-0.32$, p<0.001), but not fixed

strong (*Figure 5B*; *GMM*, $\chi^2(1)$=–0.009, p=0.93). In contrast, pre-response beta predicts RT in only the fixed strong condition (*Figure 5C*; *GMM*, $\chi^2(1)$=2.5, p=0.02 vs. weak: p>0.75 and mixed strong: $\chi^2(1)$=–0.05, p=0.52). Thus, while excursion predicted RT in all conditions, this relationship with RT was expressed differently in the absolute baseline and pre-response levels in the fixed strong context, potential reasons for which are discussed below.

## Discussion

In order to optimise detection task performance, which requires balancing true (signal-driven) against false (noise-driven) detections, the decision process must be strategically adjusted according to various task factors including one's knowledge of the evidence strength compared to the noise. In classic SDT, this is accomplished by adjusting a decision criterion which the observer uses to categorise a unitary observation as either signal or noise, setting it lower (more liberal) when targets are known to be weak. However, the normative SDT framework is based on a unitary sensory observation. In continuous detection scenarios, where fast responses are required to unpredictable targets within ongoing noisy information, decisions benefit from accumulating evidence over time. An obvious analogous mechanism for setting liberal/conservative policies in this case is to adjust a criterion on cumulative evidence, in the form of a response-triggering decision bound, as has been found to regulate speed-accuracy trade-off in discrete forced-choice decisions (*Churchland et al., 2008*; *Hanks et al., 2014*; *Murphy et al., 2016*; *Palmer et al., 2005*; *Reddi and Carpenter, 2000*; *Reddi et al., 2003*; *Thura and Cisek, 2016*). Here, we tested this prediction through the joint analysis and modelling of behavioural and electrophysiological data in a continuous motion detection task performed in three different difficulty contexts. Although we confirmed that context-dependent behaviour can be explained by a bound-adjustment model, we found that beta activity over motor cortex, a well-established neural signature of a thresholded decision variable, showed adjustment trends in the opposite direction. Beta excursion from baseline to motor execution threshold was smaller in the fixed strong context, indicative of a more liberal bound in this condition (see *Figure 2A and B*). Using the exact urgency dynamics reflected in beta oscillations which dictate the decision variable's starting level relative to threshold and hence the decision bound, we showed that behaviour can be captured equally well by a continuous accumulation model that employs an adjustable sensory criterion to which momentary evidence is referenced before accumulation.

Building on other recent studies using neurally informed modelling (reviewed in *O'Connell and Kelly, 2021*), the current study highlights the value in incorporating neural signatures of decision formation in models to uncover effects that cannot be resolved by considering behaviour alone. Specifically, the discrepancy between the behaviour and beta excursion proved to be crucial to test alternative computational mechanisms to implement context adjustments. The essential characteristic of the sensory criterion-adjustment mechanism is that it controls the strength of sensory input that is sufficient to give rise to significant positive accumulation. Within the framework of the current study, in which we assumed a common basis of continuous accumulation for all models, we implemented the criterion as a reference point subtracted from all evidence samples as they enter a continuous accumulator with lower reflecting bound. However, future research should systematically explore other viable implementations of the sensory criterion mechanism. In particular, accumulation need not be continuous and could instead be triggered, for example by the evidence first reaching a 'gate' level as has been proposed for visual search decisions (*Purcell et al., 2010*; *Purcell et al., 2012*; *Schall et al., 2011*), or by a distinct, lower-level process functioning as a relevant transition detector, neural correlates of which have been reported in humans (*Loughnane et al., 2016*) and are also present in the current data (see *Figure 2—figure supplement 2*). These gate-based triggering mechanisms provide alternative ways to implement an adjustable criterion to regulate the input to the accumulator in order to set decision policy, and could potentially be teased apart through more complex manipulations of noise and target evidence dynamics.

As we have cautioned previously (*O'Connell and Kelly, 2021*), this neurally informed modelling approach comes with the caveat that the validity of model inferences heavily depends on that of the 'linking propositions' (*Schall, 2019*) specifying how the measured neural decision signals relate to the underlying computational decision process. In this study we relied primarily on the assumption that beta amplitude, relative to its pre-response threshold level (e.g. excursion), maps onto a motor-level

decision variable driven by the additive combination of evidence accumulation and urgency components. While several previous studies support this, it is important to survey the supporting results in this specific dataset, especially given the divergence of results from the bound adjustment that we had predicted. First, beta builds over time during targets at a rate that scales with evidence strength, pointing to the evidence accumulation component. Second, beta excursion strongly predicts RT in the way predicted of a bounded accumulator with starting point variability (*de Lange et al., 2013*; *Donner et al., 2009*; *Gould et al., 2012*). Third, it also builds during the ITI where we would expect that the accumulation of noise would flat-line or quickly asymptote, and the build-up trajectory matched the increasing false alarm rates, pointing to an evidence-independent urgency component. Last, false alarms are associated with a decrease of beta reaching a threshold level closely matching that of hits, indicating a stable threshold both before and during target evidence.

When analysing absolute beta amplitudes, however, there was one result that did not align fully with previous studies. Previous studies found that beta activity usually reaches a stereotyped pre-response amplitude contralateral to the response across different conditions and RTs, suggesting a fixed threshold applied to motor preparation (*O'Connell et al., 2012*; *Steinemann et al., 2018*; *Kelly et al., 2021*). While this holds true for most conditions in the current data, pre-response beta activity in the fixed strong condition was significantly elevated relative to the other conditions (see *Figure 5A*). We side-stepped this issue in our models by constraining them using the excursion relative to the pre-response level in a given condition rather than the absolute baseline activity. However, follow-up analysis of absolute beta levels yields further insight into how the context-dependent adjustments are physiologically implemented. Although the smaller beta excursion in the fixed strong condition might suggest that people maintained a higher state of continuous preparedness, which is presumably energy-consuming (*Gottsdanker, 1975*; *Näätänen, 1972*) and thus counter-intuitive for such an easy task, the elevated absolute beta amplitude in the ITI suggests much lower tonic preparation in this context. Meanwhile, the yet greater elevation in absolute beta at the time of response could be interpreted as a more relaxed level of response caution at the motor level, which in our criterion-adjustment model is counteracted by the higher sensory-level criterion to create an overall more conservative policy. We also found that pre-response, absolute beta amplitude in the fixed strong condition was higher for faster RTs than for slower RTs. However, relative to the mixed strong condition, this elevation for the fastest trials was also observed in the baseline period (*Figure 5B*), suggesting that other, slower-fluctuating generators of beta activity that also relate to RT may super-impose on the motor preparation signal. Indeed, beta activity has been linked to many cognitive factors other than motor preparation (*Aoki et al., 2001*; *Brovelli et al., 2004*; *Buschman and Miller, 2007*; *Buschman and Miller, 2010*; *Engel and Fries, 2010*; *Murthy and Fetz, 1992*; *Siegel et al., 2012*; *Spitzer and Haegens, 2017*), and here the topographies of condition differences suggest context-dependent changes across a wider network (*Figure 5A*). This again underlines the importance of using beta excursion as model constraints to minimise the influence of slower-fluctuating interfering beta generators, but also highlights the need for future research into how other factors such as attention, response vigour, and sensorimotor connectivity can influence pre-response contralateral beta amplitudes.

An advantage of having characterised multiple neural signatures of decision formation is that one can be used to constrain models while the other can be used for additional follow-up validation. However, in these data the observed CPP and beta signals were not precisely aligned in their respective estimates of decision-bound settings. Specifically, the pre-response amplitude of the CPP (*Figure 2E*) failed to mirror the reduction in beta excursion seen in the fixed strong condition (*Figure 2B*). Under the assumption that the CPP reflects the pure cumulative evidence component that adds to urgency at the downstream motor level where the threshold is set, it should reach lower pre-response amplitudes for conditions in which beta excursion is smaller (*Steinemann et al., 2018*; *Kelly et al., 2021*). However, simulations of both neurally informed models (see *Figure 3—figure supplement 1C and D*) confirm that the predicted reduction in pre-response accumulator amplitude for the fixed strong relative to mixed strong is very small (6%), and not an effect easily detected in event-related potentials, suggesting the possibility of a type 2 error. Nevertheless, it is possible that the true empirical CPP amplitude in the fixed strong condition exceeds the value predicted in the models. We could speculate that this could arise from a relaxation of either the weighting or the delay of the transmission of CPP information into motor preparation in this easy condition, either of which

would cause the amplitudes to be higher in the fixed pre-response window without changes to model structure or constraints.

The sensory criterion adjustment mechanism presents an interesting alternative means of policy setting with respect to the strategic adjustment of the action-triggering bound suggested by the dominant computational models and empirical research on the speed-accuracy trade-off (*Bogacz et al., 2010*; *Palmer et al., 2005*). It is important to note, however, that it is not yet clear why the policy is set by sensory criterion setting here and by decision-bound setting in the speed/accuracy work, as there are two major differences: here the decision requires an adjustment for sensory target strength, not speed/accuracy emphasis, and here the task is detection within a continuous stimulus as opposed to discrimination within discrete trials. To tell which factor explains the difference in adjustment mechanism, it will be necessary to examine speed-accuracy trade-off manipulations in a continuous detection task.

In conclusion, while previous studies of decision policy adjustments for other perceptual task manipulations have found that they are enacted by changing the decision bound, we here show that, when adjusting for expected difficulty level during continuous detection, people instead regulate the transfer of sensory evidence into the accumulator process. This suggests that the brain harbours a broader range of mechanisms to regulate decision policy than it had previously seemed.

## Materials and methods
### Participants
Fourteen participants (aged 21–60 years; eight female) were recruited at University College Dublin. Two participants were left-handed, all participants had normal or corrected-to-normal visual acuity, and had no previously diagnosed psychiatric disorders, epilepsy or suffered from any head injury resulting in loss of consciousness. All participants gave written consent prior to their participation and were compensated for their time with €25. The UCD Human Research Ethics Committee for Life Sciences approved all experimental procedures in accordance with the Declaration of Helsinki (LS-16-76-Craddock).

### Experimental procedure
Participants performed the task in a dark room, while seated 57 cm from the monitor. All stimuli were presented on a black background on a 21-inch CRT monitor operating at 60 Hz refresh rate (1280×960 pixels). Stimuli were generated using custom-made software in MatlabR2016 (Mathworks, Inc, Natick, MA, USA) utilising Psychtoolbox (*Brainard, 1997*; *Kleiner et al., 2007*; *Pelli, 1997*).

#### Experimental task: RDM
Participants performed a continuous detection version of the random dot motion (RDM) task (*Hanks et al., 2006*; *Roitman and Shadlen, 2002*; *Ditterich, 2006*; *Shadlen and Newsome, 1996*). In this task, they were asked to continuously monitor a cloud of white, randomly moving dots for intermittent targets defined as a step change from random to coherent upwards dot motion lasting for 1 s (see *Figure 1A*). During periods of incoherent motion (0% motion coherence), all dots were randomly displaced to a new location throughout the patch on each frame. Coherent dot motion was accomplished by displaying a certain percentage of randomly selected dots in a direction relative to their previous location within each frame. The ITI, in which motion coherence was held at 0%, was pseudo-randomly varied among four possibilities: 2, 4, 6, or 8 s. This ITI variability created temporal uncertainty to ensure continuous engagement in the detection task. Three target-strength contexts were run in separate blocks: in the 'weak evidence context' all targets had 25% coherence; in the 'strong evidence context' all targets were 70%; and both coherence levels could appear with equal likelihood in a 'mixed context'. Participants were asked to respond to coherent upwards motion by clicking a mouse button with the index finger of their dominant hand. Each participant performed 12 blocks each consisting of 24 targets. Three blocks of each of the 'fixed' (single-coherence targets) contexts ('strong' or 'weak') were first run in an order counterbalanced across participants. The 'mixed' target coherence context was originally designed as a control condition to verify bound invariance across target strengths within a single context, and thus all participants performed six 'mixed' blocks at the end of the experimental session.

Prior to the experiment, participants completed four training blocks, each consisting of six targets with decreasing levels of coherence (90%, 70%, 45%, 25%). In the last training block, participants would practise the mixed context condition (25% and 70%). Feedback of the participant's performance was presented in text at the end of each block (numbers of hits, misses, and false alarms). The RDM pattern consists of a patch of 150 white, randomly moving dots (dot size: 4 pixels, dot speed of 3.33 deg/s) centrally presented in an aperture of 8 degree diameter. Dots were flickering on and off at a rate of 15 Hz. While this flicker was included as standard at the time to measure steady-state visual evoked potentials (see also *Kelly and O'Connell, 2013*), these potentials do not reflect sensory evidence for the decision (which is instead the motion coherence) and therefore were not analysed here.

## Data analysis

### Behavioural analysis

All trials were sorted and analysed according to the four conditions of fixed weak, fixed strong, mixed weak, and mixed strong. RT was calculated in milliseconds (ms) relative to the evidence (i.e. coherent dot motion) onset. Hits were defined as trials in which a response was made between 200 ms after target onset up to 650 ms after target offset. It was assumed that responses with RT faster than 200 ms were false alarms as it is too soon for them to be informed by the sensory evidence from the target. Meanwhile, late responses made up to 650 ms after target offset were defined as hits because this was the point where the post-target response probability returned to the level of false alarm rate during the ITI. Conversely, false alarms were defined as responses in the period from 650 ms after target offset to 200 ms after the following target onset. False alarm rate was calculated as the proportion of 2 s periods of ITI time in which such a false alarm response was made. We chose this 'per 2 s' scaling as it is the shortest ITI but note that no results are dependent on this choice of scaling, as it merely sets the units of measurement. In one exception, to look at the distribution through the ITI (*Figure 3A*, *upper*), we computed the proportion of false alarms per 1 s for each context.

Effects of the different conditions on RT were statistically analysed using a repeated-measures ANOVA (rANOVA) that organised the four conditions into two binary factors of difficulty context (fixed vs. mixed), and coherence (25% vs 70%), and with the additional factor of ITI duration preceding a given target (2, 4, 6, or 8 s). GMM were used to analyse hit rate and false alarms rate, which allows statistical testing of proportions and bounded variables. A logit link function was used for hit rates as they are binomially distributed. False alarm rates were analysed as a function of context (weak, strong, and mixed contexts). A log-link function is used for the false alarm rates as participants had unbounded opportunities to make false alarms within each ITI, yielding a Poisson distribution. For all models, intercepts and slopes were added as random effects.

### EEG acquisition and preprocessing

Continuous EEG was recorded from 128 scalp electrodes using a BioSemi system with a sample rate of 512 Hz. All data were analysed in Matlab R2018a utilising EEGLAB routines (*Delorme and Makeig, 2004*). Eye movements were recorded with four electro-oculogram (EOG) electrodes, two above and below the left eye and two at the outer canthus of each for vertical (vEOG) and horizontal (hEOG) eye movements and blinks. EEG data were low-pass filtered by convolution with a 137-tap hanning windowed sinc function to give a 3 dB corner frequency of 37 Hz with strong attenuation at the mains frequency (50 Hz; *Kelly et al., 2021*; *Widmann et al., 2015*). Noisy channels were automatically detected using the PREP pipeline (*Bigdely-Shamlo et al., 2015*) and interpolated using spherical spline interpolation. Initially, large target epochs were extracted using a window capturing the preceding ITI (up to 650 ms after the previous target offset) to 1 s after the current target offset, and shorter target epochs were further extracted from these.

Target trials were rejected if the absolute difference between the vEOG or hEOG electrodes exceeded 200 μV during –100 ms before target onset up to 100 ms after response or if 10% of the electrodes exceeded 100 μV in the target epoch. If less than 10% of the electrodes exceeded 80 μV on a given trial, then these electrodes were interpolated using a spherical spline interpolation. One participant was excluded due to excessive trial rejection (>70% in each condition) due to a mix of blinking, eye movements, and EEG artefacts. After artefact rejection, the data were subjected to a

current source density (CSD) transformation utilising the CSD toolbox in MatlabR2018a (*Kayser and Tenke, 2006*; *Kayser, 2009*).

## Decision signal analysis

We analysed two neurophysiological signals previously established to reflect decision formation. First, motor preparation was measured via effector-selective spectral amplitude decreases in the beta (15–30 Hz) frequency range. We omitted the 15 Hz frequency bin to avoid contamination of motor preparation signals by the 15 Hz steady-state visually evoked potentials (SSVEP) evoked by the on-off flicker of the dots at the same frequency. Second, we analysed the CPP, a motor-independent signal exhibiting evidence accumulation dynamics (*O'Connell et al., 2012*; *Kelly and O'Connell, 2013*). To measure beta amplitude, we computed a short time Fourier transform by applying a Fourier transform in a sliding window of 266 ms (four full cycles of the SSVEP) in steps of 10 ms, and then extracted amplitude in the 15–30 Hz frequency range for the electrode at standard sensorimotor sites C3 and C4, for participants that responded with right and left hand, respectively.

The CPP was measured from three centro-parietal electrodes that we selected from each individual participant with the highest signal-to-noise ratio (amplitude –133 to –66 ms before the response divided by the standard measurement error as suggested by *Luck et al., 2021*), within a cluster of nine electrodes delineated based on the visual inspection of the grand-average topography of the response CPP (see topography in *Figure 2D*). Beta and CPP signals were then segmented into target-locked and response-locked epochs. Target-locked epochs were extracted from –200 to 1000 ms around target onset, and response-locked epochs were extracted from –400 to 200 ms around response time. For the CPP, epochs were baseline-corrected to –133 to –66 ms around target onset (two full cycles of the SSVEP at 15 Hz). They were additionally low-passed filtered at 6 Hz with a fourth-order Butterworth filter, for plotting purposes only. Only hits, where no false alarm occurred within 1.2 s pre-target onset, were included in the analysis of stimulus- and response-locked CPP and beta.

To determine whether the neural signals show signs of decision-bound adjustment, we compared the beta amplitude 'excursion' – that is, amplitude just prior to response relative to just prior to target onset – and the pre-response CPP amplitude across the four conditions. Beta excursion was calculated on each trial by taking the difference of mean baseline amplitude (–133 to –66 ms relative to target onset) minus mean pre-response amplitude (–133 to –66 ms relative to response). CPP amplitude was calculated for each trial as the average amplitude in the pre-response time range –133 to –66 ms relative to response. The effects on beta excursion and CPP were examined using a GMM with the factors of difficulty context (fixed vs. mixed) and motion coherence (25% vs 70%), and including RT (z-scored within participant) as a covariate. For all models, random intercepts and slopes were added as random effects.

In our framework we assume that beta reflects the combination of evidence accumulation and urgency components. As the noise during the ITI is zero mean, we assumed any beta dynamics or effects in the ITI mainly reflected the operation of urgency. Thus, in order to create a standardised urgency signal to constrain computational decision models (see below), we took the grand-average beta amplitude for each context in the pre-response amplitude (–67 to –33 ms) as the threshold level for that context and subtracted it from the beta signals measured during the ITI. Then, we took the longest ITI (8 s) and fitted a second-order polynomial to capture the smooth urgency trend, ignoring any residual noise. The smooth urgency function was then normalised by dividing it by the maximum value during the ITI in any context, and finally it was inverted (urgency = 1-urgency). In this way, 0 corresponds to the lowest motor preparation level measured during the ITI in the grand-average beta signals of any context and 1 corresponds to the threshold level for triggering a response. During the targets, evidence accumulation adds to urgency and therefore beta cannot be used to isolate the dynamics of urgency on its own. For the sake of parsimony, we simply assumed that during targets following 2, 4, or 6 s ITIs, urgency continued on its trajectory measured from 8 s ITIs. For targets following 8 s ITIs, we assumed that urgency held steady at its pre-target level, based on the fact that beta tended to saturate during the ITI.

## Computational models

We fitted three alternative models to the behavioural data: a leaky accumulator model with context-dependent bound (henceforth the bound-adjustment model) and two neurally constrained models

with urgency quantified directly from beta signals as above, and featuring either leaky accumulation with context-dependent leak (henceforth the leak-adjustment model) or non-leaky accumulation of evidence cast relative to a sensory criterion with a lower reflecting decision bound at zero (henceforth the criterion-adjustment model). All three models follow a basic leaky accumulator model equation (*Ossmy et al., 2013*):

$$y(t) = y(t - dt) * (1 - Leak) + x(t) * Drift + \xi \tag{1}$$

where $y(t)$ is the accumulator output, $x(t)$ is a binary representation of target presence throughout the block (set to 1 during targets and 0 otherwise), which is scaled by drift rate 'Drift' according to the strength of the evidence for a given target and with added Gaussian noise, $\xi_s = \mathcal{N}(0, \sigma_s)$, altogether representing the noisy sensory evidence through the block. As indicated in the equation, the *Leak* parameter sets the degree to which the previous accumulator value ($y(t - dt)$) is scaled down relative to the incoming evidence. Time step size $dt$ was set to correspond to the refresh rate of the monitor (60 Hz or 16.67 ms). When the accumulator level ($y(t)$) exceeded the decision bound ($\theta$), it was then set to zero for a post-response pause of 1 s, on the assumption that subjects are unlikely to resume accumulation immediately after responding given the range of ITIs.

Additionally, a non-decision time accounting for the delay between samples of evidence appearing on the screen and registering as increments to the accumulation process was estimated as a free parameter, while post-commitment delays associated with motor execution were fixed at 80 ms based on a typical motor time estimated for mouse-button presses (*O'Connell et al., 2012*; *Steinemann et al., 2018*; *Kelly et al., 2021*). The apportioning of non-decision delays to before or after the accumulation to bound process does not generally impact the fitting of behavioural models, but we applied this fixed motor time here to be consistent with the measurement time window used for CPP amplitude. For the purposes of comparing the simulated and observed CPPs, rather than setting $y$=0 immediately upon threshold crossing, we allowed the simulated accumulator to continue accumulating for 80 ms beyond the point of decision commitment, after which the accumulator linearly decreased back to 0 over a period of 400 ms, mimicking the overshoot and fall of the empirical CPP after the response time. Note that model fits to behaviour are unaffected by the inclusion of this post-decision ramp-down. Further, to ensure that comparison of real and simulated CPP time courses would be minimally influenced by any misestimation of such time-shifts, we took measurements relative to the signal onset time in that analysis.

## Bound-adjustment model

A leaky integrator model (*Figure 1C*), where the decision bound was free to vary across the three contexts (weak, strong, and mixed), was fitted only to the behavioural data. This model was predicted a priori to best explain the decision process adaptations. This model fixed the momentary sensory noise parameter at a value of 0.1 ($\sigma_s$=0.1), and included three free decision-bound parameters – one for each context ($\theta_{context}$), one *leak* parameter, commonly applied to all conditions, one non-decision time parameter, and one for four *drift* rate parameters (see below), making a total of six or nine free parameters.

## Neurally constrained leak-adjustment model

Using the same leaky accumulator mechanism as above, this model (*Figure 3B*) instead had three leak parameters ($Leak_{context}$) so that leak was free to vary across the contexts (weak, strong, and mixed). Here, the decision bound was fixed to 1 and additionally the sensory noise parameter ($\sigma_s$) was a free parameter since the decision variable is now scaled by the normalised bound. Here, the sum $y(t) + u(t)$ was computed as the motor-level decision variable and subjected to the fixed decision bound to determine responses (with the non-decision time implemented as in the previous model), where $u(t)$ is the urgency function measured from observed beta signals during the ITI as detailed above in Decision signal analysis. This resulted in six free parameters for the minimal model with one free *Drift* rate parameter, and nine for the expanded model with four free *Drift* rates.

## Neurally constrained criterion-adjustment model

In a second neurally constrained model (*Figure 3C*), we considered a non-leaky accumulation of evidence that is recast as a quantity relative to a sensory-level criterion (as distinct from a *Criterion*

set on cumulative evidence in the decision variable, which we refer to as a 'decision bound'), with the additional setting of a lower reflecting bound at zero on the accumulator, according to:

$$t = max(0, y(t - dt)) + (x(t) * Drift + \xi_s - C_{\text{context}})$$ (2)

Here, sensory criterion ($C_{context}$) was free to vary across the three contexts (weak, strong, and mixed). The fixed bound, constrained urgency, and free noise parameter are exactly as in the neurally constrained leak-adjustment model above. Again, the most minimal model with a single free drift rate parameter thus has a total of six free parameters (including one non-decision time as usual), and nine free parameters for the version with four free *Drift* rates.

## Behavioural model fitting

These models were each fitted to a total of 32 behavioural data points, comprised of the proportion of all targets with a hit in six RT bins defined by the 0.1, 0.3, 0.5, 0.7, and 0.9 RT quantiles or with no response (misses) across the four target conditions, and the false alarm rate measured as a proportion of every 2 s segment of the ITIs before the targets of each condition (see above and *Figure 1B*). The models were fitted using Monte Carlo simulations of the decision process to generate simulated behavioural responses, which were compared to the observed data using the $\chi^2$-based statistic $G^2$:

$$G^2 = 2(\sum_{i=1}^{4} n_i \sum_{j=1}^{8} p_{ij} \log \frac{p_{ij}}{\pi_{ij}})$$ (3)

Here, $n_i$ is the number of valid trials, $p_{ij}$ and $\pi_{ij}$ are the observed and the simulated proportion, respectively, for condition $i$ in RT bin $j$. $G^2$ statistic was minimised with a bounded Nelder-Mead SIMPLEX algorithm (*Nelder and Mead, 1965*; *fminsearchbnd* in Matlab).

Similar to *Corbett et al., 2021*, the models were fitted in several stages: In the first stage, a minimal model was fitted that included all relevant free parameters for that model (see above) but only one free drift rate parameter for the weak coherence, which was then linearly scaled for the strong conditions according to relative coherence (70/25). In this initial broad search, about 4500 trials were simulated per model evaluation in separate SIMPLEX fits for 1000 different initial 'guess' parameter vectors. These initial guess vectors were made by sampling from a uniform distribution spanning a reasonable range along each parameter dimension, so that the guess vectors together comprehensively covered a large parameter space in which to find a global minimum. In a second refinement stage, about 20,000 trials were simulated in each model evaluation in separate SIMPLEX fits starting from the 30 best estimated parameter vectors arising from the first step, with lower tolerance criteria for termination or convergence. This second step tended to improve the $G^2$ only slightly, typically by about 2%.

Last, we ran a similar refinement step for model versions that now had four separate free drift rate parameters, to allow capturing any modulation of evidence accumulation quality due to either practice (as the participants always performed the mixed condition last) or context itself (e.g. drift rate may depend on how often strong targets arise in a given context). Using the 30 best refined parameters, we produced 200 new initial parameter vectors by adding random Gaussian jitter of a standard deviation of 1.1 times the mean of each of the refined parameters and 1.15 times of the mean for the drift rate parameters. Again about 20,000 trials were simulated in each model evaluation for this last step.

To confirm that the added complexity of four free drift rates was warranted over a single free drift rate parameter within each model, we computed the Akaike's (AIC) and Bayes information on criterion (BIC), to penalise for the complexity of the models, for example number of free parameters ($f$).

$$\text{AIC} = G^2 + 2f$$ (4)

$$\text{BIC} = G^2 + f \log(\sum_{i=1}^{4} n_i)$$ (5)

The principal comparisons in the study were across the three models with the same number of free parameters in any comparison, only differing in accumulation and/or adjustment mechanisms. Therefore, these models were simply compared using $G^2$ itself.

## Acknowledgements

SPK and ACG were supported by research grants from Founder grant number 15/CDA/3591 and The Wellcome Trust (219572/Z/19/Z). RGO was supported by Horizon 2020 European Research Council Consolidator Grant Ind Decision 865474.

## Additional information

### Competing interests

Redmond G O'Connell: Reviewing editor, *eLife*. The other authors declare that no competing interests exist.

### Funding

| Funder | Grant reference number | Author |
|---|---|---|
| Science Foundation Ireland | 15/CDA/3591 | Anna C Geuzebroek |
| Wellcome Trust | 219572/Z/19/Z | Anna C Geuzebroek |
| Horizon 2020 European Research Council Consolidator Grant Ind | 865474 | Redmond G O'Connell |

For the purpose of Open Access, the authors have applied a CC BY public copyright license to any Author Accepted Manuscript version arising from this submission. The funders had no role in study design, data collection and interpretation, or the decision to submit the work for publication.

### Author contributions

Anna C Geuzebroek, Software, Formal analysis, Validation, Visualization, Methodology, Writing – original draft; Hannah Craddock, Conceptualization, Data curation, Project administration; Redmond G O'Connell, Conceptualization, Methodology, Writing - review and editing; Simon P Kelly, Conceptualization, Software, Formal analysis, Supervision, Methodology, Writing - review and editing

### Author ORCIDs

Anna C Geuzebroek ⓘ https://orcid.org/0000-0002-8287-2990
Redmond G O'Connell ⓘ http://orcid.org/0000-0001-6949-2793
Simon P Kelly ⓘ http://orcid.org/0000-0001-9983-3595

### Ethics

All participants gave written consent prior to their participation and were compensated for their time with €25. The UCD Human Research Ethics Committee for Life Sciences approved all experimental procedures in accordance with the Declaration of Helsinki (LS-16-76-Craddock).

### Decision letter and Author response

Decision letter https://doi.org/10.7554/eLife.83025.sa1
Author response https://doi.org/10.7554/eLife.83025.sa2

## Additional files

### Supplementary files

• MDAR checklist

### Data availability

Code to recreated the Random Dot Motion task utilising Psychtoolbox is publicly available at https://github.com/AnnaCGeuzebroek/Context-Dependent-Detection (copy archived at *Geuzebroek, 2023*). All code to recreated the behavioural and EEG data analysis as well as the modelling code can be found at https://github.com/AnnaCGeuzebroek/Continuous-Behavioural-Modelling (copy

archived at *Geuzebroek, 2022*). Pre-processed anonymised EEG and behavioural data is uploaded at OSF https://osf.io/yjvku/?view_only=7ed5aee5d09a4d5ca13de1ba169b0588.

The following dataset was generated:

| Author(s) | Year | Dataset title | Dataset URL | Database and Identifier |
|---|---|---|---|---|
| Geuzebroek AC, Kelly S | 2022 | Balancing true and false detection of intermittent sensory targets by adjusting the inputs to the evidence accumulation process | https://doi.org/10.17605/OSF.IO/YJVKU | Open Science Framework, 10.17605/OSF.IO/YJVKU |

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
