## [Editor Report]

This important work reveals a novel mechanism by which perceptual decision-making is regulated as a function of task demands. The combination of behavioral and physiological (EEG) evidence supporting the accumulation of evidence referenced to a context-dependent sensory criterion is convincing. Overall, the study makes a strong case for the importance of augmenting behavioral modeling with additional input from neural signatures of the underlying decision process.

---

## [Decision Letter]

**Decision letter after peer review:**

Thank you for submitting your article "Balancing true and false detection of intermittent sensory targets by adjusting the inputs to the evidence accumulation process" for consideration by *eLife*. Your article has been reviewed by 2 peer reviewers, and the evaluation has been overseen by Valentin Wyart as Reviewing Editor and Michael Frank as the Senior Editor. The following individual involved in the review of your submission has agreed to reveal their identity: Romy Frömer (Reviewer #2).

The reviewers have discussed their reviews with one another, and the Reviewing Editor has drafted this to help you prepare a revised submission. As you will see, the two reviewers have found that your work makes a strong case as to how behavior alone can be insufficient to tease apart competing models of sequential sampling models of decision-making, and possibly lead to misattribution of observed behavioral differences. Your neurally-informed modeling approach shows how neural measures may help in such situations to arbitrate between models that are indistinguishable from behavior alone. However, both reviewers agreed that this approach relies strongly on the assumed relationship between the neural measure of interest (here, the Β amplitude signal) and the decision process, something which amounts to 'reverse' inference to some extent. It is therefore important to consider more explicitly the possibility that this assumed relationship may either be incorrect in some respect – e.g. because the neural measure reflects a mixture of components of the decision process rather than a single component. The approach is nevertheless novel, and potentially powerful, and the main text could provide all the available empirical evidence in the data that points toward the assumed relationship between Β amplitude and evidence-independent urgency.

The reviewers have agreed on the below list of essential revisions that should be addressed in a point-by-point fashion and accounted for in a revised version of your manuscript. The individual reviews of the two reviewers are also provided at the bottom of this decision letter for you to address also minor comments on the manuscript. These individual reviews do not require point-by-point responses. We hope that you will be able to address these comments and that you will choose to submit your revised manuscript to *eLife*.

Essential revisions:

1) The revised manuscript should give greater consideration to the possibility that the hypothesized relationship between the neural measure of interest (Β amplitude) and the decision process may not be entirely correct. The neurally-informed modeling approach is indeed quite appealing to distinguish between candidate models that are indistinguishable from behavior alone. However, it appears at times in the current manuscript to rely on a form of 'reverse' inference: assuming that the neural measure of interest is a faithful, selective correlate of a specific component of the decision process. It is almost never the case that such inferences can be made without significant doubt, especially in the case of non-invasive neural measures such as EEG-derived metrics.

A specific paragraph in the Discussion section would be very useful for readers to clarify not only the benefits of your novel approach but also its assumptions and potential caveats.

2) The neurally-informed modeling approach (Figure 3) appears to use Β amplitude as an evidence-independent urgency signal that adds to the decision variable (it is described as such in the Methods section). However, Β amplitude is clearly affected by the evidence (Figure 2A). And in the Results section, Β amplitude is described as best corresponding to a decision variable, which would include a dependence on sensory evidence. This seems conflicting: how can Β amplitude be used as an evidence-independent urgency signal in the neurally-informed modeling if it depends on sensory evidence?

To resolve this apparent discrepancy between the assumption of the neurally-informed modeling approach and the data, it would be very important to provide the key properties of the Β amplitude signal (ideally from your own dataset) that make it suitable to use as an evidence-independent urgency signal. This list should ideally be found in the Results section so as to give readers a clear idea of the several properties of this signal before using it as an evidence-independent urgency signal in the modeling. Because the interpretability of the results of these analyses depends on this assumption, it is key that the reader understands how much this modeling assumption is grounded in the available data. Nevertheless, a Discussion paragraph that explains how the neurally informed modeling approach depends on this assumption (see point 1) is very important.

3) Related to the previous point, the properties of the Β amplitude signal suggest that it reflects a mixture of processes rather than a selective component of the decision process. As noted by Reviewer 2, in the evidence criterion adjustment model, the average DV value during the ITI will be closer to the decision bound in the weaker signal condition compared to the stronger signal condition (this is how the model fits the false alarm rate differences). This should be reflected in Β amplitude if the latter reflects the DV. However, the Β-derived urgency is highest for the condition with the lowest false alarm rate and lowest for the condition with the highest false alarm rate. It would be very useful to clarify these seemingly hypothesis-inconsistent aspects of the data in the revised manuscript, which may be resolved with further details and explanations regarding what is expected of the neural signal of interest.

4) Regarding the criterion-adjustment model, could there be an alternative equivalent account that does not involve regulation of the transfer of incoming evidence? It appears highly similar to one with a constant negative drift added to the decision variable in addition to the contribution of evidence (and a lower reflecting bound).

5) Parameterization of the models. You explain clearly that one of the model parameters (e.g., noise or bound) must be fixed, but it is not clear why you didn't keep it the same parameter across all models. Couldn't you have fixed the noise parameters for all the models? Having it different across models makes it difficult to compare parameter values across the models, especially because the fitted noise term in the neurally-informed models appears dramatically reduced compared to the fixed value of noise used for the bound-adjustment model. Also regarding model parameterization, could you possibly report the leak parameter using more intuitive units? It seems to be parameterized as a fraction leak per time step, but the time step is unclear. Reporting the leak as a time constant would be immediately understandable.

6) Reviewer 2 has identified several problems with Figure 3, which should be corrected in the revised version: Figure 3 could benefit from a number of improvements. The various line types in the top panel of A are not labeled or described. The range of values of the y-axis in the top panel seems rather small compared to the overall false alarm rates. I know the latter was calculated as a proportion over 2-second intervals. I don't know the bin width for Figure 3A, but the numbers being a factor of 10 smaller seems pretty far off based on my best guess of bin width. The scaling of the bottom panel of A clips off 2 of the 3 curves at earlier times. From the methods, I believe at least one of the curves should go all the way down to 1, so half the range is cut off in the current version. In panel B, the evidence distributions are not labeled and a bit confusing. I can guess that they correspond to distributions with a strong signal, with a weak signal, and without a signal. However, the presumptive "without signal" black distribution appears shifted negative of zero, which I don't think is correct. The title of panel F says "simulated Bound-adjustment" while the text describes this as the "simulated criterion-adjustment" model. I presume this is a mistake in labeling the title, but as this is the most critical distinction for the paper's main conclusion, it's really important to get it right.

7) In Figure 3F (simulation of the criterion-adjustment model), it is indeed quite puzzling that the DV is at zero at the time of target onset. You explain that the urgency signal was removed from the simulations of this "DV", which is useful, but there should still be an influence of pre-target noise. Did the simulations not include any influence of pre-target noise? Reviewer 2 is correct that it is unclear why it is not included, and it should be explored explicitly if possible.

8) It would be useful to discuss at least some alternative models that may have not been considered in this study. Modeling studies and their comparison are always dependent on the models being included. One could maybe have thought of a different alternative to the standard model, such as the precision-weighted evidence integration mentioned by Reviewer 3. It would be useful to discuss alternative variants of the standard account of the contrast between conditions somewhere in the Discussion section.

Frömer, R., Callaway, F., Griffiths, T., & Shenhav, A. (2022, October 22). Considering what we know and what we don't know: Expectations and confidence guide value integration in value-based decision-making. Retrieved from psyarxiv.com/2sqyt

*Reviewer #1 (Recommendations for the authors):*

In addition to the comments in the Public Review, Figure 3 could benefit from a number of improvements. The various line types in the top panel of A are not labeled or described. The range of values of the y-axis in the top panel seems rather small compared to the overall false alarm rates. I know the latter was calculated as a proportion over 2-second intervals. I don't know the bin width for Figure 3A, but the numbers being a factor of 10 smaller seems pretty far off based on my best guess of bin width. The scaling of the bottom panel of A clips off 2 of the 3 curves at earlier times. From the methods, I believe at least one of the curves should go all the way down to 1, so half the range is cut off in the current version. In panel B, the evidence distributions are not labeled and are a bit confusing. I can guess that they correspond to distributions with a strong signal, with a weak signal, and without a signal. However, the presumptive "without signal" black distribution appears shifted negative of zero, which I don't think is correct. The title of panel F says "simulated Bound-adjustment" while the text describes this as the "simulated criterion-adjustment" model. I presume this is a mistake in labeling the title, but as this is the most critical distinction for the paper's main conclusion, it's really important to get it right. To underscore that point, if this is in fact the simulation of the criterion-adjustment model, I'm surprised that the DV is at zero at the time of target onset. I realize that the urgency signal was removed from the simulations of this "DV", but there should still be an influence of pre-target noise. For the other models, the noise-driven DV averages to zero, but with the zero floor of the criterion-adjustment model, it shouldn't peg 0 exactly during this time. I suspect that the simulations did not include any influence of pre-target noise, and I would suggest that is something that should be included.

*Reviewer #2 (Recommendations for the authors):*

This paper was a pleasure to read. I found it very clearly written, with a clear rationale and thorough tests of the theoretical predictions. The methods are impressive but conveyed in a way as to not scare off readers.

I would love to be helpful, but I can't see where the authors could use my advice.

Perhaps out of curiosity: Instead of a sensory criterion as implemented by the authors, could the same pattern be achieved with precision-weighted evidence integration? (cf. Frömer, R., Callaway, F., Griffiths, T., & Shenhav, A. (2022, October 22). Considering what we know and what we don't know: Expectations and confidence guide value integration in value-based decision-making. Retrieved from psyarxiv.com/2sqyt).

---

## [Author Response]

Essential revisions:1) The revised manuscript should give greater consideration to the possibility that the hypothesized relationship between the neural measure of interest (Β amplitude) and the decision process may not be entirely correct. The neurally-informed modeling approach is indeed quite appealing to distinguish between candidate models that are indistinguishable from behavior alone. However, it appears at times in the current manuscript to rely on a form of 'reverse' inference: assuming that the neural measure of interest is a faithful, selective correlate of a specific component of the decision process. It is almost never the case that such inferences can be made without significant doubt, especially in the case of non-invasive neural measures such as EEG-derived metrics.A specific paragraph in the Discussion section would be very useful for readers to clarify not only the benefits of your novel approach but also its assumptions and potential caveats.

We fully agree that it is very important to question and discuss the assumptions upon which our modelling approach relies especially as the results are contradicting our initial hypothesis. As suggested by the reviewers, we have now added a new paragraph in the Discussion section listing the pieces of evidence within this dataset supporting the link between Β oscillations relative to pre-response levels and the motor-level decision variable (see line 343-359, page 11 and 12), while also highlighting the caveat that inferences can only ever be as accurate as these linking propositions. In the process, we also decided to show more of the empirical evidence in figures. In particular, revised Figure 2B now shows the prediction of reaction time by Β excursion, and Figure 2 —figure supplement 1 shows the Β waveforms for false alarms, verifying there is a decrease to a similar threshold as for true target detections.

2) The neurally-informed modeling approach (Figure 3) appears to use Β amplitude as an evidence-independent urgency signal that adds to the decision variable (it is described as such in the Methods section). However, Β amplitude is clearly affected by the evidence (Figure 2A). And in the Results section, Β amplitude is described as best corresponding to a decision variable, which would include a dependence on sensory evidence. This seems conflicting: how can Β amplitude be used as an evidence-independent urgency signal in the neurally-informed modeling if it depends on sensory evidence?To resolve this apparent discrepancy between the assumption of the neurally-informed modeling approach and the data, it would be very important to provide the key properties of the Β amplitude signal (ideally from your own dataset) that make it suitable to use as an evidence-independent urgency signal. This list should ideally be found in the Results section so as to give readers a clear idea of the several properties of this signal before using it as an evidence-independent urgency signal in the modeling. Because the interpretability of the results of these analyses depends on this assumption, it is key that the reader understands how much this modeling assumption is grounded in the available data. Nevertheless, a Discussion paragraph that explains how the neurally informed modeling approach depends on this assumption (see point 1) is very important.

We thank the reviewers for this comment as we indeed were not consistently clear on our assumptions regarding Β oscillations. As suggested, we clarified provided a list of properties in the Results section (see line 111-125, page 4) and added a cautionary discussion in the Discussion section (see line 343-359, page 11 and 12). Furthermore, we now provide more Β analyses using our own dataset to examine this critical issue (see line 133-140, page 6). First, reviewer 1 is correct that Β reflects both urgency and evidence accumulation, which we now state more clearly, and this underpins our assumption that Β amplitude reflects a thresholded decision variable additively combining urgency and cumulative evidence. For clarity we now exclusively use the term “decision variable” in reference to this motor-level, thresholded process combining both components. Our assumption, following Kelly et al. (2021), was that during the inter-target interval (ITI) when the evidence is zero-mean, we can take the cumulative evidence component as negligible and use Β excursion during the ITI as a pure measurement of urgency. However, as pointed out in Comment 3, this assumption does not hold for the criterion-adjustment model and so we describe below a new control model fit addressing this.

We have made edits in the Introduction, Methods, Results and Discussion that are consistently clear on our *‘Β = evidence accumulation + urgency’* assumption, and we additionally added a list of properties supporting this assumption from the present dataset in the Results section and reiterate them in the Discussion:

Β builds at an evidence-dependent rate as decisions are formed during targets, as clearly indicated in Figure 2A,

To add a fourth property, we characterise the Β decreases just prior to false Alarms. Initially we did not include any false alarm waveforms as there were almost no false alarms in the Fixed strong conditions. Nevertheless, Β waveforms for false alarms are still illustrative for the other conditions and we have included them in the supplementary material (Figure 2 —figure supplement 1). Two aspects are important: first that false alarms are associated with a local Β decrease, reaching the same pre-response Β amplitude for false alarms as the pre-hit level, supporting the assumption in the models that the same threshold level applies through the ITI as during the target epochs (Figure 2 —figure supplement 1B). Second, there is a post-response increase or β “rebound” effect as typically observed (Pfurtscheller and Silva, 1999). Although the decreasing and increasing phases appear to roughly balance so that this effect in the minority of ITIs containing false alarms is unlikely to impact the measurements of static offsets between urgency signals during the ITI, to be conservative, we have replotted and recalculated all target-locked and response-locked waveforms in Figure 2A to exclude trials where a false alarm occurred within the 1.2 seconds prior to the target onset. These new figures still show the same pattern as in the initial manuscript and confirms that the occurrence of false Alarms is not artificially elevated baseline Β in the Weak condition away from the decision threshold.

3) Related to the previous point, the properties of the Β amplitude signal suggest that it reflects a mixture of processes rather than a selective component of the decision process. As noted by Reviewer 2, in the evidence criterion adjustment model, the average DV value during the ITI will be closer to the decision bound in the weaker signal condition compared to the stronger signal condition (this is how the model fits the false alarm rate differences). This should be reflected in Β amplitude if the latter reflects the DV. However, the Β-derived urgency is highest for the condition with the lowest false alarm rate and lowest for the condition with the highest false alarm rate. It would be very useful to clarify these seemingly hypothesis-inconsistent aspects of the data in the revised manuscript, which may be resolved with further details and explanations regarding what is expected of the neural signal of interest.

We thank the reviewer for raising this counter-intuitive aspect of the criterion-adjustment model. As mentioned in response to Comment 2, we do assume that Β excursion represents the additive combination of cumulative evidence plus the urgency signal. For the leaky accumulation models the cumulative noise in the ITI has a trial-average of zero, allowing us to assume that during the ITI in the absence of evidence that the average Β reflects the urgency component only. In contrast, as reviewer 1 points out, in the criterion-adjustment model, due to the reflective bounds, the average asymptotic level of noise accumulation would be higher for the Weak condition due to the lower sensory criterion feeding more positive values to the accumulator. Illustrating this, we simulate the individual urgency and evidence accumulation components, along with the total DV reflected in Β, arising from the original fit of the criterion-adjustment model (Author response image 1). This highlights the inaccuracy in our original assumption that Β is ‘evidence independent’ during the ITI. We had estimated the urgency offsets directly from Β measurements, when Β actually reflects the sum of the urgency component plus this context-dependent average level of noise accumulation during the ITI. The simulation shows that the inaccuracy is not large – the differences in tonic levels of noise accumulation (Author response image 1) are small compared to the assumed urgency offsets (Author response image 1), and so the aggregate motor-level DV offsets (Author response image 1) are still in the direction observed, with Strong elevated over Weak, albeit smaller. Nevertheless, it was important to correct for this inaccuracy in a control analysis to verify that the overall conclusions do not change.

**Author response image 1. sa2fig1:** Simulations of the original criterion-adjustment model showing the traces throughout the longest ITI (8 sec). Here showing, (A) the urgency constrained by the Β data, (B) “evidence” (zero-mean noise during ITI) accumulation and (C) the sum of the urgency component plus evidence accumulation*.* These figures show that within the original criterion-adjustment model the noise accumulates to a context-dependent average asymptotic level, which contributes alongside urgency to the context-dependent offsets seen in the motor-level decision variable reflecting the sum urgency + evidence accumulation.

In the control analysis, we refitted the data to not only reproduce the observed behavioural patterns but also, simultaneously, the static shifts observed in baseline Β amplitude in Figure 2C. Specifically, we now constrained the time-dependent urgency function to equal the quadratic approximation of Β during the ITI for the Mixed condition only, and added two free parameters to capture the static shifts in urgency for the Weak and Strong conditions, relative to the Mixed. The models were fitted using Monte Carlo simulations of the decision process to generate simulated behavioural responses as well as the normalised Β excursion, whose match to the observed data was evaluated by adding a penalty term *w*(observed Β – simulated Β)^2^* to the G^2^ value in the objective function to be minimised (Described in Lines 222 – 238, page 8). This gave us the results that are now added in Table 2 (page 8) and Figure 3 —figure supplement 1.

‘In the above neurally-constrained model fits, we assumed that the cumulative evidence component of motor preparation is negligible during the ITI so that Β excursion directly constrain the urgency component only. A potential issue with this arises from the fact that the accumulator operates continuously in our models: whereas the zero-mean noise in the ITI would accumulate to an average level of zero in the leaky accumulation model, the lower reflecting bound in the criterion-adjustment model would cause a positive asymptotic value of the average evidence accumulation during the ITI which scales inversely with criterion. This would predict highest tonic activity in the Weak context as it has the lowest criterion, so it does not constitute a viable alternative explanation the observed lowest Β offset in the Weak context. Nevertheless, it highlights a likely inaccuracy in the model’s estimation of the degree of adjustment of criterion and urgency offsets. We therefore carried out a revised model fitting procedure in which urgency offsets were not set directly by Β excursion, but rather free to vary in order to reproduce the observed static shifts in the Β excursion across contexts simultaneously with the behavioural data see detailed explanation in Figure 3 —figure supplement 1. The behavioural fits still favoured the criterion-adjustment model (G2 + penalty = 24) over the leak-adjustment model (G2 + penalty = 33; see Table 2), and as expected, estimated even larger urgency offsets to compensate for the asymptotic evidence accumulation levels.’

These new model fits show that even when fixing this issue of ignoring noise accumulation differences, the criterion-adjustment model is able to capture the behaviour while adopting the counterintuitive urgency offsets that result in the static shift in pre-target motor preparation in which the Strong condition is closest to the bound but has the least false alarms. The results of this control analysis have been added to Table 2. In Author response image 2 we show a re-simulation of the individual urgency and evidence accumulation components and the sum reflecting the predicted motor-level decision variable for this control analysis, in the same way as above. Here we see that the urgency offsets lie even more strongly in the originally observed direction (Author response image 2), more than counteracting the smaller offsets in the noise accumulation, so that the total decision variable now more closely matches the offsets in the observed Β traces (see Author response image 2).

**Author response image 2. sa2fig2:** Simulation from the control analysis where the Criterion-adjustment model was forced to reproduce the static shifts observed in baseline Β amplitude as well as the observed behavioural patterns simultaneously. Traces in the Longest ITI are plotted, from left to right: (A) Urgency traces, where the Mixed condition is constrained by the observed Β signals and the Weak and Strong conditions are statically shifted by two free parameters relative to the Mixed condition; (B) the evidence (zero-mean noise during ITI) accumulation; (C) the sum of the urgency component and evidence accumulation.

While this control analysis quantitatively reconciles the apparent discrepancy in the context-dependent effects in Urgency versus average noise accumulation during the ITI, it remains quite counter-intuitive that the observed Β, in the reviewers’ words, “is highest for the condition with the lowest false alarm rate and lowest for the condition with the highest false alarm rate.” We therefore provided an additional simulation of the decision variable’s behaviour at the single-trial level (taking a random single trial for illustration), which demonstrates that since the accumulator spends much of its time at the reflecting bound level of zero during the ITI, the total decision variable spends most of its time lying lower for the Weak than the Strong context, as dictated by the dominant urgency offsets; however, due to the lower criterion set in the Weak context, it more often happens that a short sequence of noise samples drives the accumulator high enough to accidentally trigger a false alarm in that context. This is shown in Figure 3 —figure supplement 2.

4) Regarding the criterion-adjustment model, could there be an alternative equivalent account that does not involve regulation of the transfer of incoming evidence? It appears highly similar to one with a constant negative drift added to the decision variable in addition to the contribution of evidence (and a lower reflecting bound).

This is true – a constant negative drift rate would work also – but as reviewer 1 pointed out, this is equivalent to the criterion adjustment model. In fact, this is exactly what the criterion-adjustment model does, subtracting a constant value from all the evidence samples before being accumulated. We refer to it as “referencing to a criterion” but this subtraction is the same as adding a negative drift. We refer to this equivalence when we mention the “drift criterion” of Ratcliff. Our Discussion also lays out other possible implementations of the regulation of incoming evidence, for example involving triggered rather than continuous accumulation. These would still adhere to the claims of the paper.

5) Parameterization of the models. You explain clearly that one of the model parameters (e.g., noise or bound) must be fixed, but it is not clear why you didn't keep it the same parameter across all models. Couldn't you have fixed the noise parameters for all the models? Having it different across models makes it difficult to compare parameter values across the models, especially because the fitted noise term in the neurally-informed models appears dramatically reduced compared to the fixed value of noise used for the bound-adjustment model. Also regarding model parameterization, could you possibly report the leak parameter using more intuitive units? It seems to be parameterized as a fraction leak per time step, but the time step is unclear. Reporting the leak as a time constant would be immediately understandable.

Thank you for pointing out the potential difficulty in evaluating parameter values from the table. To avoid confusion when comparing parameter values between the neurally-informed and bound-adjustment models, we have added an explanation for these differences in scaling parameters in the table caption. For the bound-adjustment model, we made noise σ = 0.1 the fixed scaling parameter both because this is the most common practice in the diffusion model literature and because the bound parameter needed to be free to vary across the three contexts. In the neurally-constrained models it is the bound itself that provides the most natural anchorpoint because all urgency estimates are measured relative to the pre-response amplitude that is taken to correspond to that bound. We considered rescaling the bound-adjustment model to facilitate direct quantitative comparison in the table, but this would entail some arbitrary choices that in turn may lead to confusion; for example, a natural rescaling would be to take the bound in one of the conditions – say, the Mixed condition – to be equal to 1 in the bound-adjustment model just as it is in all conditions in the neurally-constrained models, and dividing all other bound parameters and the drift rate parameters by 0.85. This would require adding another column for the bound-adjustment model to enter the rescaled value of the noise (0.1/0.85), which would make it less clear at a glance what free parameters were in each model. What’s more, because the neurally-constrained models have a strong urgency component the effective bound on cumulative evidence is much less than 1 and varies not only across conditions but also across ITIs, further complicating any attempt we might make to try to bring the bound-adjustment model into register for direct comparison. We hope, therefore, that the reviewer will agree that adding the below to the Table 1 caption is on balance the least confusing way to guide the reader in how to compare parameter values across models, though of course we are happy to consider other suggestions the reviewer may have. For the leak parameter, we also added an explanation to the caption to avoid confusion, and provided the time step size as suggested by the reviewer. Since we speak in terms of “more and less leak” in the text rather than “shorter or longer time-constant,” we opted to stick to the leak parameter, but with a clear explanation as to how to interpret it.

See the new Table 1 caption, page 4: ‘Parameter values estimated for the leaky accumulator model with adjustable bound fit to behaviour alone, and for the neurally-informed models featuring leak-adjustment and criterion-adjustment. ’Noise’ refers to the standard deviation (σ) for the Gaussian sensory evidence noise. ’Bound’ refers to the threshold set on the decision variable for triggering a response. ’Leak’ refers to the proportion of the current cumulative evidence total that leaks away on the following sample (note a 16.7 ms time step is used). No leak was fitted for the criterion-adjustment model. ’Tnd’ refers to the non-decision time in ms. ’Drift’ refers to drift rate, corresponding to the mean of the sensory evidence during targets. The Goodness-of-Fit metric, G2, is listed for each model. Note that, if comparing parameter values between models directly, it must be taken into account that whereas the bound-adjustment model set a scaling parameter of noise σ = 0.1 and allowed bounds to vary freely with respect to this. In contrast, the neurally-constrained models were scaled directly by the normalised urgency signals relative to the ultimate action triggering bound taken to be equal to 1. These fixed parameters are indicated in red.’

6) Reviewer 2 has identified several problems with Figure 3, which should be corrected in the revised version: Figure 3 could benefit from a number of improvements. The various line types in the top panel of A are not labeled or described. The range of values of the y-axis in the top panel seems rather small compared to the overall false alarm rates. I know the latter was calculated as a proportion over 2-second intervals. I don't know the bin width for Figure 3A, but the numbers being a factor of 10 smaller seems pretty far off based on my best guess of bin width. The scaling of the bottom panel of A clips off 2 of the 3 curves at earlier times. From the methods, I believe at least one of the curves should go all the way down to 1, so half the range is cut off in the current version. In panel B, the evidence distributions are not labeled and a bit confusing. I can guess that they correspond to distributions with a strong signal, with a weak signal, and without a signal. However, the presumptive "without signal" black distribution appears shifted negative of zero, which I don't think is correct. The title of panel F says "simulated Bound-adjustment" while the text describes this as the "simulated criterion-adjustment" model. I presume this is a mistake in labeling the title, but as this is the most critical distinction for the paper's main conclusion, it's really important to get it right.

We thank the reviewer for catching each of these issues. In addition to correcting the mistakes, since Figure 3 contained a lot of information, we decided that the manuscript would benefit from splitting this figure into two: (1) Figure 3 which includes the false alarm and motor preparation trends during the ITI and the two Neurally-informed models with the behavioural data, and (2) Figure 4 which shows the validation analysis based on the simulated evidence accumulation traces and their first derivatives. This also allows us to add a similar simulation of the Bound-adjustment model, which illustrates that the level of leak in that behaviour-only model predicts an even starker immediate drop in buildup rate. We now address each of the figure issues in turn:

The various line types in the top panel of A are not labelled or described.

We have now added a legend for Figure 3A; the different-coloured lines refer to the different context conditions, and for each, the four ITIs are superimposed. Since across ITIs these false Alarm trends are repeats of the same underlying function, we now show all in solid lines, avoiding the distraction of different line styles. We now explain this clearly in the caption on page 7.

The range of values of the y-axis in the top panel seems rather small compared to the overall false alarm rates. I know the latter was calculated as a proportion over 2-second intervals. I don't know the bin width for Figure 3A, but the numbers being a factor of 10 smaller seems pretty far off based on my best guess of bin width.

We thank the reviewer for catching this error in proportion calculation in Figure 3A (upper) – it is now corrected and we now state that it is per 1-second (the bin width in this plot) rather than per 2-sec period.

The scaling of the bottom panel of A clips off 2 of the 3 curves at earlier times. From the methods, I believe at least one of the curves should go all the way down to 1, so half the range is cut off in the current version.

In Figure 3A (lower) the y-axis is actually still in units of Β amplitude relative to the pre-response level (the same as Figure 2A), i.e. the signal has not been normalised yet. We did indeed zoom in just slightly, to more clearly distinguish the relevant differences between the three context conditions. We think that our original inversion of the axis so that “up” corresponds to Β decrease may have caused this confusion, and so we have flipped the y-axis back to normal, as in Figure 2A, with the pre-response level corresponding to zero now at the bottom, and fixed the caption to clarify that Β as shown has not yet been normalised to scale between 0 and 1.

In panel B, the evidence distributions are not labeled and a bit confusing. I can guess that they correspond to distributions with a strong signal, with a weak signal, and without a signal. However, the presumptive "without signal" black distribution appears shifted negative of zero, which I don't think is correct.

We thank the reviewer for catching this; we have now corrected it so that the criterion point “C_0_” in the leak-adjustment model is fixed at zero, the very centre of the noise distribution.

The title of panel F says "simulated Bound-adjustment" while the text describes this as the "simulated criterion-adjustment" model. I presume this is a mistake in labeling the title, but as this is the most critical distinction for the paper's main conclusion, it's really important to get it right.

Thank you – we have checked the titles and corrected them.

7) In Figure 3F (simulation of the criterion-adjustment model), it is indeed quite puzzling that the DV is at zero at the time of target onset. You explain that the urgency signal was removed from the simulations of this "DV", which is useful, but there should still be an influence of pre-target noise. Did the simulations not include any influence of pre-target noise? Reviewer 2 is correct that it is unclear why it is not included, and it should be explored explicitly if possible.

Thank you for noting this. We do indeed simulate with pre-target noise, but what we now make clear is that we baseline-correct this data in the same way as the observed CPP, i.e. -133 to -66 ms relative to target-onset, to be able to compare the simulated CPP with the empirical data. Note, that in the models this correction is not applied and in the simulation there is indeed an effect of pre-target noise, as we showed above in response to Comment 3. Note also, as we stated above, that for clarity we now avoid using the term “DV” for the accumulation component, and reserve the use of “decision variable” or “DV” to refer to the TOTAL motor-level process subjected to threshold.

8) It would be useful to discuss at least some alternative models that may have not been considered in this study. Modeling studies and their comparison are always dependent on the models being included. One could maybe have thought of a different alternative to the standard model, such as the precision-weighted evidence integration mentioned by Reviewer 3. It would be useful to discuss alternative variants of the standard account of the contrast between conditions somewhere in the Discussion section.Frömer, R., Callaway, F., Griffiths, T., & Shenhav, A. (2022, October 22). Considering what we know and what we don't know: Expectations and confidence guide value integration in value-based decision-making. Retrieved from psyarxiv.com/2sqyt

We thank the reviewer for this. In our Discussion we describe some alternative model implementations but do not provide fits here because they would not change the central claim that unlike speed-accuracy tradeoff, the strategic policy adjustments to take account of target strength are not the direct bound adjustments. In ongoing work we are testing alternative models on this and other continuous monitoring datasets mostly focussing on continuous vs. alternative gating models, but the results are orthogonal to this claim. In other work in our group, we are also examining continuous motion direction identification around 360 degrees, which will lend itself to examining aspects of precision.